# Adherence to public institutions that foster cooperation

Arunas L. Radzvilavicius[1,2], Taylor A. Kessinger[1,2] & Joshua B. Plotkin [1✉]

Humans typically consider altruism a moral good and condition their social behavior on the moral reputations of others. Indirect reciprocity explains how social norms and reputations support cooperation: individuals cooperate with others who are considered good. Indirect reciprocity works when an institution monitors and publicly broadcasts moral reputations. Here we develop a theory of adherence to public monitoring in societies where individuals are, at first, independently responsible for evaluating the reputations of their peers. Using a mathematical model, we show that adherence to an institution of moral assessment can evolve and promote cooperation under four different social norms, including norms that previous studies found to perform poorly. We determine how an institution's size and its degree of tolerance towards anti-social behavior affect the rate of cooperation. Public monitoring serves to eliminate disagreements about reputations, which increases cooperation and payoffs, so that adherence evolves by social contagion and remains robust against displacement.

[1] Department of Biology, University of Pennsylvania, Philadelphia, PA, USA. [2]These authors contributed equally: Arunas L. Radzvilavicius, Taylor A. Kessinger.
✉email: jplotkin@sas.upenn.edu

Social norms, reputations, and institutions of moral assessment are critical to cooperation in human societies[1–6]. People typically consider altruistic behavior a moral good[7], and they tend to be more cooperative when observed by others[8,9]. Moral reputations are directly related to cooperation, and cooperation in turn can lead to higher social status[10], while kindness towards people of bad moral standing is sometimes punished[11]. Interactions in modern societies often involve cooperation with strangers, and so people must rely on moral reputation scores provided by institutions or third parties, for instance, in e-commerce interactions[5,12]. Historically, communities have constructed institutions that broadcast information about others' behavior so that all reputations are publicly available[13,14].

Evolutionary game theory provides a convenient framework to study human behavior governed by social norms, reputations, and community enforcement[15]. In such models of indirect reciprocity, a donor's action (to cooperate or not) depends on the recipient's moral reputation ("good" or "bad"). Reputations are updated according to a social norm – a collection of rules that prescribe how an individual's reputation depends on her past behavior towards others[16,17]. A simple social norm called Stern Judging, for example, assigns a good reputation to those who refuse to help individuals of bad moral standing and to those who cooperate with other good members of the society[3,18].

According to the large literature on indirect reciprocity, cooperation can evolve and remain stable under simple social norms like Stern Judging, as individuals adopt discriminatory behavior and cooperate only with players of good reputation[15,18–20]. However, this explanation for cooperation relies on a public reputation system or rapid gossip[15,21–23], so there are no disagreements about the reputations of people among their peers. But in many realistic settings, people make their own subjective moral judgments about one another, and their views may differ due to different observation histories or independent errors in observing a focal individual's actions[24]. Under a framework of private reputations, even when everyone follows the same norm of moral assessment, cooperation tends to collapse due to disagreements in the population about each others' moral standings[25,26].

One way to restore cooperation under private moral assessment is by empathetic perspective taking[27,28]. Although the reputation of a focal individual may vary according to the perspective of different observers (Fig. 1a), empathy can reduce the rate of misunderstandings and unjustified defection. Moreover, empathy itself can evolve through social contagion, inducing high rates of cooperation typical of societies that enjoy an established public monitoring system. Nonetheless, empathetic perspective taking is not always a feasible solution to the problem of cooperation under private moral evaluations. Evaluating social interactions from the perspective of another person carries a high cognitive cost, and not everyone is capable of empathizing or perspective-taking[29]. Moreover, inferences about the perspective of another person are not always accurate, and the benefits of empathy are vulnerable to the possibility of deception and manipulation[30]. For instance, a potential donor wishing to avoid the cost of cooperation with a recipient without being assigned a bad reputation could falsely portray a bad subjective assessment of all of her potential partners.

Aside from empathy, we hypothesize that cooperative behavior can also arise when moral evaluations are delegated to an institutional observer, or observers, who broadcast public reputation scores for all members of the society. We define a such a public institution as a sub-population of $Q$ individuals chosen to broadcast their consensus view on the reputations of everyone in the population (Fig. 1b). An institution can consist of a single individual, $Q = 1$, the whole society, $Q = N$, or any intermediate number of observers. The institution's consensus view of a focal individual is determined by the mean reputation from the perspectives of the observers within the institution, along with a fixed strictness threshold $q$. In particular, the consensus public reputation of a focal individual is broadcast as "good" provided a proportion $q$ or more of the observers who form the institution see the focal individual as good. The threshold $q$ is thus a measure of institutional strictness towards antisocial behavior. Under a strict institution (high $q$) even occasional antisocial behavior results in being assigned a bad public reputation, whereas under a tolerant institution (low $q$) occasional antisocial behavior is permitted without assigning a bad public reputation.

Human societies have developed a variety of cultural institutions that reinforce conditional cooperation[31]. Here, we extend the theory of indirect reciprocity to study the difference between private moral assessment and institutionalized, public moral evaluation. We first investigate how adherence to different types of institutions for moral evaluation determines equilibrium rates of cooperation, as behavioral strategies are allowed to evolve.

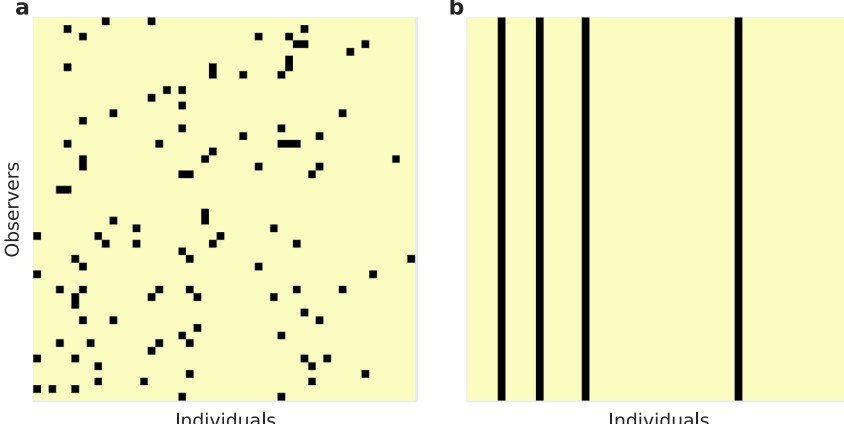

**Fig. 1 Reputation matrices showing good (yellow) and bad (black) reputations of individuals (horizontal axis) as seen from the perspectives of different observers (vertical axis). a** When moral evaluations are private, observers may disagree on an individual's reputation. Disagreements arise from independent errors in observation or because different observers assess different social interactions involving a focal individual. **b** Instead of performing moral evaluations independently, people may instead choose to rely on reputations provided by a centralized institution, which in this figure consists of $Q = 2$ distinguished observers (first two rows) who broadcast their consensus views. When the entire population relies on the public institution for moral assessment, reputational disagreements are eliminated and cooperation can be facilitated.

Next we consider the evolution of adherence itself to a public institution of assessment. We allow individuals either to rely on reputation scores from their own subjective moral assessments of others or, alternatively, to trust the reputation scores provided by an extant institution. Depending upon the size of an institution and the degree to which it tolerates antisocial behavior, we find that adherence to the public institution can spread by social contagion. We conclude that adherence to public institutions of moral assessment can spread naturally, and we determine the conditions under which institutions produce high rates of cooperation for each of the four social norms we study. Both parts of our study – strategic evolution assuming institutional adherence, and the emergence of adherence itself – rely on mathematical analysis by replicator dynamics for two-player games[32,33] as well as Monte Carlo simulations.

## Results

**Model.** We study pairwise donation games with a payoff matrix of the form $\begin{pmatrix} b-c & -c \\ b & 0 \end{pmatrix}$. In any given interaction, if the donor chooses to cooperate with the recipient, she pays a cost $c$ and her interaction partner receives a benefit $b > c$. However, if the donor defects, she pays no cost and the recipient receives no benefit.

Individuals make decisions based on their behavioral strategy. We consider three strategies: ALLC (always cooperate), ALLD (always defect), or DISC (cooperate only with recipients of good reputation). The frequencies of these three strategies in the population evolve via social contagion – that is, individuals tend to copy the strategies of more successful players (see Methods Section "Strategy evolution under public monitoring"). Reputations are updated according to a social norm: a set of rules defining what types of behavior result in the donor being assigned a good or bad reputation when observed by a third party[16,34]. While these rules can in general be complex, we focus on second-order normative rules that depend only on the donor's action and the recipient's reputation, since these norms tend to outperform more complex norms[17].

Specifically, we study the four social norms most prominent in the literature on indirect reciprocity: Stern Judging $\begin{pmatrix} G & B \\ B & G \end{pmatrix}$, Simple Standing $\begin{pmatrix} G & B \\ G & G \end{pmatrix}$, Scoring $\begin{pmatrix} B & B \\ G & G \end{pmatrix}$, and Shunning $\begin{pmatrix} B & B \\ B & G \end{pmatrix}$. The row index $i$ indicates the donor's action, $i = 1$ for defect or $i = 2$ for cooperate, and the column index $j$ indicates the reputation of the recipient, $j = 1$ for bad or $j = 2$ for good. All four of these norms endorse (with a good reputation) an individual who cooperates with a morally good recipient and condemn (with a bad reputation) an individual who defects against a morally good recipient, but they differ in how they treat interactions with morally bad recipients. Simple Standing assigns a good reputation for any interaction with a bad recipient, whereas Shunning assigns a bad reputation for any interaction with a bad recipient. Stern Judging endorses defection with bad recipients and condemns cooperation with bad recipients. Finally, Scoring endorses cooperation with bad recipients and condemns defection with bad recipients, so that in this case reputational assessments depend only on the action of the donor, not on the reputation of the recipient.

Our model also includes errors in strategy execution and in observation[35]: an individual who intends to cooperate can accidentally defect with probability $e_1$, and an observer may erroneously assign a bad reputation instead of a good reputation, and vice versa, with probability $e_2$.

We consider societies of individuals who are initially responsible for their own private moral evaluations of observed interactions: their evaluation of the donor depends on the donor's action they observe, the social norm, and the recipient's reputation. These evaluations are characterized by a level of empathy $0 \le E \le 1$: with probability $1 - E$, an observer will evaluate the donor using her own view of the recipient's reputation, but with probability $E$, the observer will empathize with the donor, using instead the donor's view of the recipient's reputation[27]. Alternatively, individuals can make their behavioral decisions using the public reputation of the potential recipient, as provided by a monitoring institution. The monitoring institution consists of $1 \le Q \le N$ distinguished observers who broadcast their consensus view of each person's reputation. The public reputation of an individual is broadcast as "good" only if more than $qQ$ institutional observers view her as good; otherwise, it is broadcast as "bad". Here, $0 \le q \le 1$ is the strictness threshold of the institution, which quantifies the institution's degree of tolerance towards antisocial behavior: a lower value of $q$ corresponds to a more permissive institution. First we study the effect of different institutions on levels of cooperation, and then we turn to the problem of whether adherence to a public institution will spread in a population.

**Cooperation under institutionalized moral assessment.** We first investigate how different types of institutions for moral assessment influence cooperative behavior, assuming complete adherence to the public broadcast. The broad consensus in the literature on indirect reciprocity is that Stern Judging is the most socially beneficial norm of moral assessment, as it promotes the most cooperation among the second-order norms[17,18,20,27]. In the hierarchy of norms, Stern Judging is closely followed by Simple Standing, whereas Shunning and Scoring are generally incapable of supporting cooperation rates exceeding 50%[20,35]. These prior results on social norms and cooperation assume a standard model of public information about reputations, which corresponds to a special case of our model: an institution with a single member $Q = 1$.

In contrast to prior studies, we find that, regardless of the social norm adopted by the society, the size and strictness of a public institution of moral assessment can be specified to support cooperation at rates as high as, or even higher than, achieved under Stern Judging.

To establish these results, we use a mathematical model based on replicator dynamics[32,33] to describe changes over time in the frequencies of the three strategies (ALLC, ALLD and DISC) in an infinite population. Strategies spread or decline according to their payoffs relative to the population mean (see Methods Section "Strategy evolution under public monitoring"). We assume that reputation assessments equilibrate before individuals copy strategies of their more successful peers. That is, the timescales of reputation dynamics are faster than those of strategic change[35]. We contrast the strategic dynamics in a society that follows a public institution to one in which individuals each form private moral judgments[26,27]. Initially, we consider two extreme types of institutions: very strict institutions, which broadcast an individual's reputation as good only if all members agree she is good ($q > (Q-1)/Q$), and very tolerant institutions, which broadcast an individual's reputation as good provided at least one member views her as good ($q < 1/Q$).

For all four second-order social norms considered, we find that public institutions of moral assessment generally support two strategic equilibria: a monomorphic population of unconditional defectors or a population consisting solely of discriminators (Fig. 2). To quantify the likelihood that cooperation will evolve in

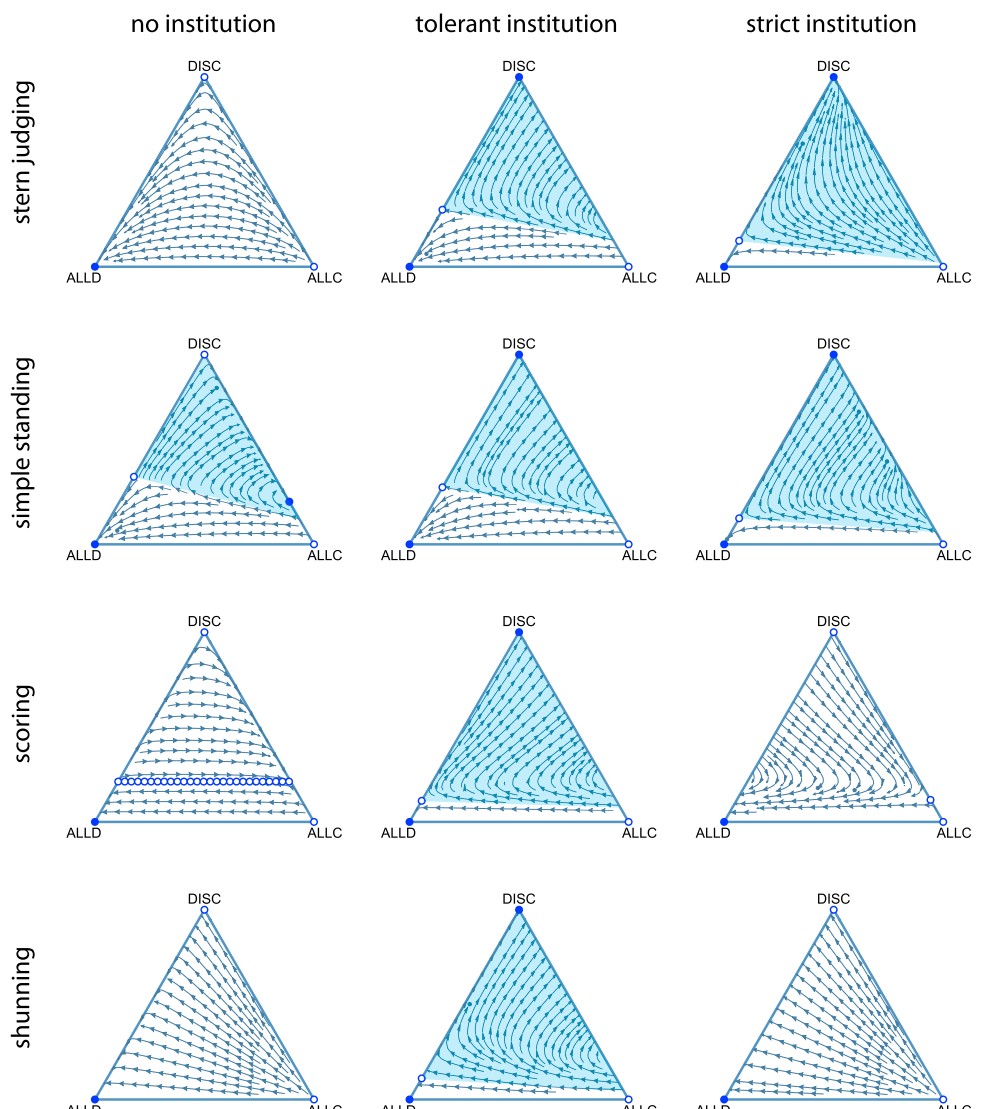

**Fig. 2 Public institutions of moral evaluation can facilitate the evolution of cooperation.** We analyzed the evolution of behaviors in the donation game for societies with unempathetic private moral assessment (left column) compared to societies with two types of institutionalized moral judgment: tolerant ($q < \frac{1}{2}$, middle column) or strict ($q > \frac{1}{2}$, right column). The triangular simplex represents the frequencies of three behavioral strategies in an infinite population: ALLC (always cooperate), ALLD (always defect), and DISC (cooperate with good individuals, defect against bad), for each of four different social norms (rows). Blue arrows indicate the direction of strategic evolution under replicator dynamics, with circles indicating stable (filled) and unstable (empty) equilibria. The basins of attraction towards stable equilibria that support cooperation are highlighted in light blue. Error rates are $e_1 = e_2 = 0.02$, benefits and costs are $b = 5$, $c = 1$, and institution size is $Q = 2$. Analogous results for $e_1 \ll e_2$, $e_1 \gg e_2$, and $b = 10$ are shown in Supplementary Figs. 10, 11, and 12.

such scenarios, we investigate the basins of attraction of cooperative equilibria, that is, the set of initial strategic states that lead to the dominance of discriminators and corresponding high levels of cooperation.

The basin of attraction towards the cooperative equilibrium – that is, the proportion of initial conditions that lead to cooperation[33] – depends on the social norm and on the institution's degree of tolerance towards antisocial behavior (Fig. 2). Under the Scoring and Shunning norms, there are large basins of attraction toward cooperation when public reputations are provisioned by highly tolerant institutions, whereas neither strict institutional assessment nor private assessment support DISC as a stable equilibrium at all. Under Stern Judging and Simple Standing, both strict and tolerant public institutions support substantial basins of attraction towards cooperation, but the basins are even larger for strict institutions. Stern Judging,

unlike Simple Standing, does not support any stable cooperation under (non-empathetic) private assessment.

Supplementary Figs. 10, 11, and 12 demonstrate how the basin of attraction towards cooperative play depends upon model parameters: the error rates $e_1$ and $e_2$, as well as the benefit-to-cost ratio $b/c$. Varying error rates such that either $e_1 \gg e_2$ or $e_1 \ll e_2$ has nearly imperceptible effects on the basins of attraction for all the social norms we study. By contrast, increasing the benefit-to-cost ratio $b/c$ by a factor of two has a noticeable quantitative, but not qualitative, effect: it increases the size of any non-empty basin of attraction towards cooperation.

Mathematical analysis via replicator dynamics allows us to understand the dynamics of behavior and equilibrium cooperation rates in the limit of large population size (Fig. 2). The evolution of strategies in small societies is additionally subject to demographic stochasticity and random strategy exploration at a

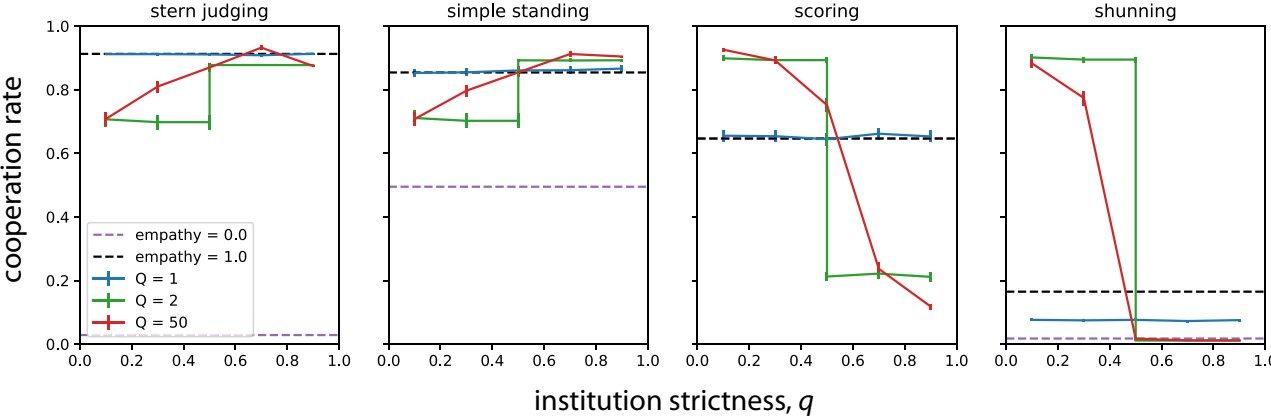

**Fig. 3 Public institutions for moral assessment govern the level of cooperation as behavioral strategies are allowed to evolve.** For each social norm, it is possible to specify a public institution of appropriate size ($Q$) and strictness ($q$) that induces far higher levels of stationary mean cooperation (solid lines) than in societies with private moral assessment (dashed lines). Public institutions can outperform even empathetic private assessment. The public institution that maximizes cooperation always requires more than $Q = 1$ individuals. Strict institutions (high $q$) perform best under Stern Judging and Simple Standing, whereas tolerant institutions (low $q$) perform best under Scoring and Shunning. Lines indicate the mean cooperation rate across 2500 replicate Monte Carlo simulations with population size $N = 50$, mutation rate $\mu = 0.025$, error rates $e_1 = e_2 = 0.02$, and benefit and cost of cooperation $b = 5$ and $c = 1$. Error bars indicate 95% confidence intervals of the mean ($\pm 2$ standard errors).

rate $\mu$. To study cooperation in this setting, we performed Monte Carlo simulations in which successful strategies spread via social contagion. That is, a randomly chosen individual $i$ copies the strategy of another randomly chosen individual $j$ with a probability $1/(1 + \exp(-w[\Pi_j - \Pi_i]))$, where $w$ is the selection strength and $\Pi_i$ and $\Pi_j$ are the payoffs of $i$ and $j$ averaged over all their interactions[36,37]. As in[27], we assumed that each individual plays many rounds of the donation game, with different partners, before reputations are updated and strategy imitation takes place (see Methods Section "Monte Carlo simulations" for simulation details).

Monte Carlo simulations show that public institutions of moral assessment can support high levels of cooperation in finite populations (Fig. 3). As predicted by our analysis of replicator dynamics (Fig. 2), the stationary mean rate of cooperation in a finite population depends on both the institution size $Q$ and its strictness threshold $q$ towards antisocial behavior. Importantly, for every social norm we study, we find that an institution of appropriate size and strictness can support high levels of cooperation. By contrast, under the classic model of public information about reputations, which corresponds to an institution of size $Q = 1$, high rates of cooperation could occur under only Stern Judging or Simple Standing[20,35]. The public institution that maximizes cooperation for a given norm always requires more than $Q = 1$ members, although size $Q = 2$ is typically sufficient to achieve as much cooperation as possible under yet larger institutions.

As predicted by our mathematical analysis of replicator dynamics (Fig. 2), simulations in finite populations governed by Stern Judging show that rates of cooperation are highest for strict institutions, which engender as much cooperation as empathetic societies using private moral assessment (Fig. 3). Under the Simple Standing norm, we also find that strict institutional assessment performs best, and it sustains cooperation at levels that even exceed those under empathetic private assessment. By contrast, forgiving institutions produce less cooperation under Simple Standing or Stern Judging, although those institutions still outperform unempathetic private assessment (Fig. 3).

Scoring is the only norm of moral evaluation that does not account for the recipient's reputation: it universally treats cooperation as a moral good. Simulations show that cooperation in societies governed by Scoring are nonetheless highly sensitive

to the type of institution. Cooperation rates are highest under tolerant institutional assessment, where they greatly exceed the rates of cooperation in populations with private moral assessment (Fig. 3). We observe a similar trend under Shunning, the norm that condemns any social interaction with individuals of bad moral character. In contrast to prior studies of public information (which corresponds to an institution of size $Q = 1$), we find that Shunning can engender cooperation levels as high as those supported by Stern Judging, provided the institution has at least two members and is sufficiently tolerant (Fig. 3).

Why do some norms require tolerant institutions of moral assessment to promote cooperation, while others require strict institutions? One possibility has to do with a norm's sensitivity to occasional antisocial behavior. Shunning and Scoring are the two norms for which a single interaction with a bad individual can trigger a cascade of punishment and defection that eventually leads to low cooperation rates. In these cases tolerant institutions help individuals avoid being assigned bad reputations from occasional interactions with bad players, and so a high frequency of cooperating discriminators is less likely to be dislodged by occasional errors, mutation, or drift under tolerant Shunning or Scoring. Under Simple Standing, on the other hand, strict institutions promote more cooperation because only those individuals who defect against good are labeled bad, and high values of $q$ help keep these defectors in check. The same mechanism holds under Stern Judging, where strict institutions are more efficient at inducing selection against willful defectors.

**Evolution of institutional adherence.** We have shown that high levels of cooperation can be achieved under second-order social norms in a society that uniformly follows an institution of public monitoring, with appropriate size and strictness. But individuals could instead assess reputations based on their own personal observations of social interactions[26]. And so a critical question remains: why would individuals choose to abandon their personal judgments in favor of trusting the reputations reported by a centralized monitoring system?

To investigate whether trust in public monitoring can evolve via individual-level selection, we studied institutional adherence as a cultural trait that can itself spread from individual to individual by copying behavior, where individuals tend to copy

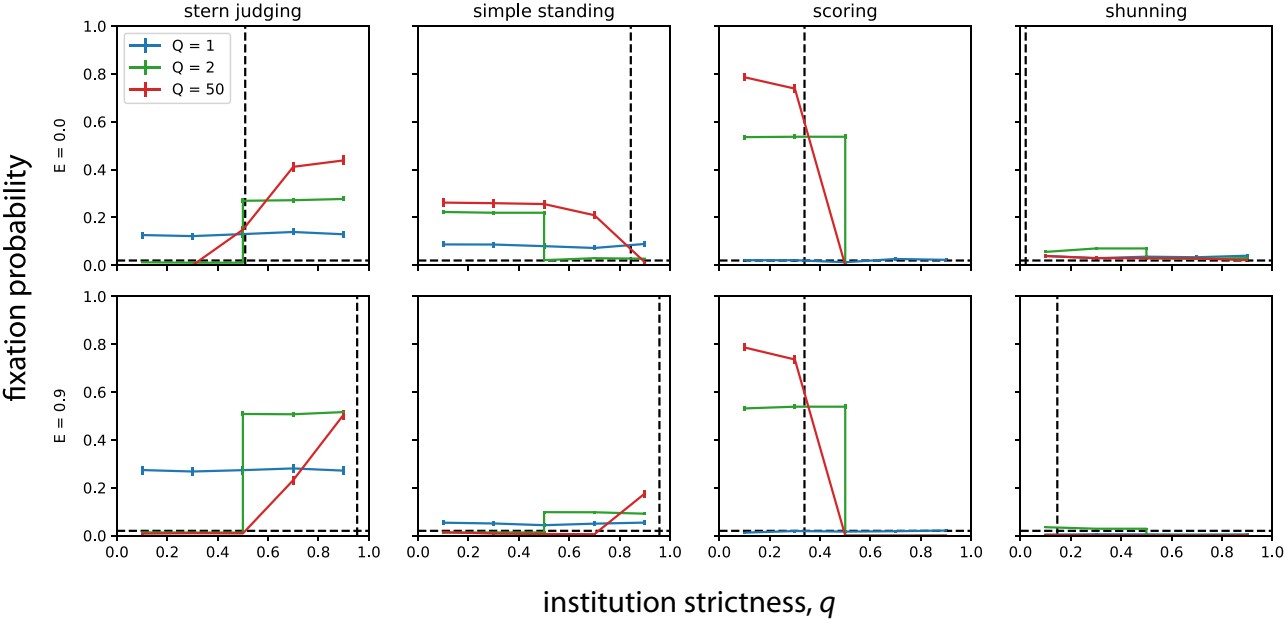

**Fig. 4 The evolution of adherence to a public institution of moral assessment.** We introduce a single institutional adherent (DISC-ADHERE) into a population consisting of non-adherents (DISC-PRIVATE), and we record the proportion of Monte Carlo simulations in which the institutional adherent eventually fixes in the population. Fixation probabilities exceeding $1/N$ indicate that adherence to the institution is favored by selection, compared to neutral drift. Vertical dotted lines represent mean frequencies of "good" reputations in the population at the time of adherent introduction, averaged over all subjective perspectives. Horizontal dotted lines indicate the neutral fixation probability, $1/N$. Error bars indicate 95% confidence intervals for the binomial fixation probability measured across 2500 replicate Monte Carlo simulations with population size $N = 50$, error rates $e_1 = e_2 = 0.02$, and the institution comprised of an external board of $Q$ members (see Methods Section "Competition between institutional adherents and non-adherents", and also Supplementary Fig. 7 for analogous results with an internal board).

the traits of others who have relatively higher payoffs. We developed replicator dynamics to quantify the selection pressure on institutional adherents competing against individuals who use private assessment (Methods Section "Competition between institutional adherents and non-adherents"). We compared these analytical predictions to Monte Carlo simulations in a finite population of DISC players, in which we tested whether a single institution-following individual can invade and overtake the population by social contagion, resulting in a population that uniformly adheres to institutional broadcasts.

We find that adherence to public institutions can often be favored by cultural evolution and eventually fix in the population. The spread of institutional adherence tends to homogenize the reputation matrix and eliminate unjustified defections that occur under private assessment of reputations[27], thereby increasing individual-level fitness. The conditions that promote public monitoring depend upon the social norm, and they also depend on the level of empathy employed by those individuals who rely on private judgments (Fig. 4).

In Stern Judging societies, adherence to a strict institution is much more likely to evolve than adherence to a tolerant institution, and adherence is favored to fix only if the ancestral population has low levels of empathetic perspective taking (Fig. 4). Likewise, for Simple Standing, highly empathetic societies are not likely to evolve adherence to public monitoring, whereas populations dominated by non-empathetic moral evaluators are likely to evolve adherence to institutions with large Q for all but the strictest values of the threshold $q$.

Under Scoring, institutions of size $Q = 1$ are neutral and hence unlikely to fix when rare. But adherence to a tolerant institution consisting of more than one observer is overwhelmingly favored by selection (Fig. 4) and will eventually engender high levels of cooperation (Fig. 3). This result is independent of empathy, since assessment by Scoring takes into account only the action of the

donor and not the reputation of her partner. On the other hand, societies with low empathy are unlikely to evolve institutional adherence under Shunning, as fixation probabilities of adherence are always close to neutral. In empathetic societies of Shunners, however, tolerant institutions are favored by selection, resulting in fixation and high rates of cooperation.

Why are some types of institutions more likely to evolve widespread adherence than others? Our findings provide some evidence for the idea that the equilibrium frequency of good individuals prior to the emergence of any institutional adherents (vertical dotted lines in Fig. 4) may be responsible for these differences. When individuals move from private to public monitoring, they move from using relative reputations to objective reputations (a positive effect): however, depending on the tolerance threshold $q$, they may lose some capacity to discriminate, which can lead to either unconditional cooperation or defection. Depending on the norm, then, those who follow an institution can sometimes be punished (assigned a bad reputation) by the majority of the society that does not yet follow the institution.

In cases where the equilibrium frequency of "good" players is initially low, such as societies governed by Scoring and Shunning, high values of the institutional threshold $q$ means that adherents will universally see all of their peers as bad individuals. As a result, an institution-following mutation will not invade, since under both of these norms an individual who defects against a bad recipient is punished, in effect, by being assigned a bad reputation. Those who do not choose to trust the institutionalized reputation system will be better able to discriminate between the members of the society who are worthy of their help and the members who are unworthy. And so only tolerant institutions, with $q$ close to the mean frequency of "good" players in the ancestral population, will be able to invade under the norms of Scoring or Shunning (Fig. 4).

On the other hand, under Stern Judging, the initial frequency of good players is intermediate (e.g., 50% with $E = 0$), and an individual who first adheres to a tolerant institution will see most players as good. As a result, the adherent will engage in a high frequency of un-reciprocated cooperation, and, in addition, she will be punished half of the time (assigned a bad reputation) by those peers who do not yet follow the institution. Because of this double cost, a small proportion of institutional adherents cannot invade. On the other hand, with high values of $q$, when the institution-following invader is rare, most of the reputations provided by the institution will be bad, especially when $Q$ is large. As a result, the invader will not cooperate, she will receive a higher-than-average payoff, and she will be assigned a bad reputation only half of the time. As the invader frequency increases, so does the frequency of good players (averaged over all perspectives), and institutional adherence will eventually fix. Similar dynamics apply to a rare institution-following invader under the Simple Standing norm, but in this case the frequency of good players is already high in the ancestral population and cooperation is rewarded from the very beginning of the invasion.

Whatever the social norm, once the trait responsible for following a public institution sweeps to fixation in a population, it is thereafter robust. That is, adherence cannot be selectively displaced by individuals who use only private moral assessment (Supplementary Fig. 14). This means that institutions of moral assessment are evolutionarily stable. In particular, analogous to our results on invasion by institutional adherents (Fig. 4), once adherents have replaced non-adherents they are thereafter robust to displacement under strict Stern Judging and tolerant Scoring, as well as Shunning and Simple Standing regardless of the institution's strictness (Supplementary Fig. 14). There is a simple intuition that explains this stability. Once the vast majority of a population follows the public institution, the rare invader who relies on her own subjective assessment is at a disadvantage because she may hold different beliefs about reputations than others hold. For instance, the rare invader will sometimes see an institutionally good individual as bad, will defect, and will subsequently be punished through reciprocal defection by the institution-following majority.

To illustrate the frequency-dependent effects summarized above, we also calculated the mean fixation probability of adherence, starting from a random initial frequency of institution-following individuals drawn uniformly from the open interval (0, 1). The mean fixation probability over this distribution of initial conditions provides a different measure of the stability of institutional adherence against those who use private reputations. The mean fixation probability depends, once again, on the type of the institution ($q$ and $Q$), the social norm, and the degree of empathy among non-adherents (Supplementary Fig. 1). Under Stern Judging and low empathy, strict institutions for public monitoring have a high fixation probability over almost the entire range of initial adherent frequencies, resulting in a high mean fixation probability. Under Simple Standing, adherence is stable across a broad range of institutional tolerances. And under Scoring and Shunning, tolerant institutions are more strongly stabilized by evolution.

The results above describe institutional adherents competing against private assessors, assuming that everyone pays attention to reputations, i.e., assuming DISC players. We have also explored the fate of discriminators who adhere to public broadcasts (henceforth DISC-ADHERE) when competing against more diverse alternatives: discriminators using private assessment (DISC-PRIVATE), unconditional defectors (ALLD), and unconditional cooperators (ALLC). Even when introduced into a population containing these three other types, institutional adherents are often favored by selection, although their fixation probability may be reduced (Supplementary Figs. 3–6), especially when ALLD is frequent. Still, for some norms and institutions, such as strict Stern Judging, selection can promote adherence even in a population initially dominated by ALLD (Supplementary Fig. 5) or by ALLC (Supplementary Fig. 6).

**Crowding effects of empathy and institutions**. Whether evolution promotes adherence to a public institution of judgment depends on whether private assessors have an internal capacity for empathy. We have seen that adherence is less likely to spread in a population of empathetic private assessors, as opposed to egocentric assessors (Fig. 4). This result makes intuitive sense, because empathy allows populations to achieve high levels of cooperation even in the absence of a public information[27], and so there is less marginal benefit for empathetic individuals to adhere to a public institution.

These results mirror the effect known as motivational crowding, which has broad theoretical and empirical support in economics[38–40]: external interventions can undermine intrinsic motivations. It is interesting to note, however, that the general pattern of interference between intrinsic empathy and extrinsic institutions is not ubiquitous. Figure 4 reveals two counter-examples: first, adherence to a strict public institution can selectively invade a population under the Stern Judging norm, whether or not the initial population is empathetic; and second, adherence to a large tolerant institution can invade under the Shunning norm, even when the private assessors are empathetic. (Note that in the case of the Scoring norm, empathy cannot possibly interfere with institutional adherence, because this social norm does not account for the recipient's reputation when assessing a donor.) The extent to which these two counter-examples coincide with the few identifiable cases of synergy between external and internal incentives in the economics literature[40] remains a topic for further study.

**Replicator dynamics of institutional adherence**. We have used Monte Carlo simulations to study whether institutional adherence will spread in a finite population (Fig. 4). But these simulations do not provide a systematic account of how adherence depends on parameters, including error rates ($e_1$ and $e_2$), the costs and benefits of cooperation ($b$ and $c$), institution size ($Q$), and institution strictness ($q$). To address these questions analytically, we developed replicator dynamic equations describing competition between institutional adherents and private assessors in an infinite population (Methods Section "Competition between institutional adherents and non-adherents"). These replicator dynamics are more complicated than those that describe strategy competition among adherents alone, because in this setting two individuals may disagree about a third individual's reputation.

We have derived replicator dynamics to describe competition among four types of individuals: unconditional cooperators (ALLC), unconditional defectors (ALLD), discriminators who adhere to the public institution (DISC-ADHERE), and discriminators who rely on their own private assessments (DISC-PRIVATE). This derivation requires that we separately track the frequency of individuals within each of these four types that is seen as good either by private assessors or by the public institution, as well as the frequency of disagreements both among private assessors and between private assessors and public adherents. Finally, we must specify whether the $Q$ members of the institution are themselves engaged in strategic interactions ("internal board", Methods Section "Competition between institutional adherents and non-adherents"), or whether the institutional assessors are separate from the population ("external board").

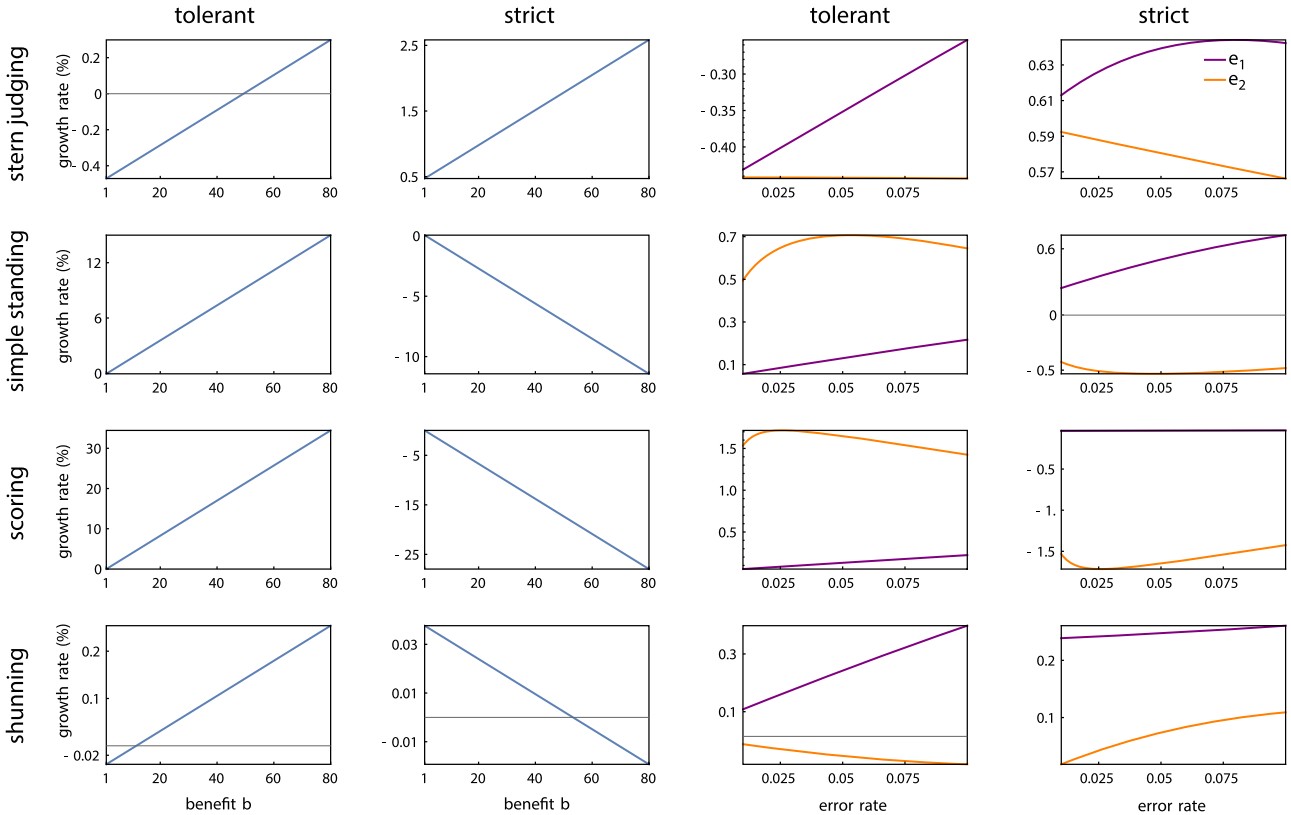

**Fig. 5 Growth rate of institutional adherents (DISC-ADHERE) in a population otherwise composed of private assessors (DISC-PRIVATE).** The instantaneous growth rate per unit time is computed according to replicator dynamics in an infinite population (Methods Eq. (11)) with the initial frequency of adherents set at 1/50, for strict ($q > 1/2$) or tolerant ($q < 1/2$) institutions of size $Q = 2$ (external board). Varying the benefit-to-cost ratio can have a qualitative impact on the evolution of institutional adherence. For example, under tolerant Stern Judging, adherents are strongly suppressed by selection for $b/c < 50$ but favored by selection for $b/c > 50$. Unless otherwise varied in a figure panel, parameter values are $e_1 = e_2 = 0.02$, $b = 5$, $c = 1$.

The resulting replicator dynamics describe how all model parameters – including the error rates, payoff matrix, and size and strictness of the public institution – influence dynamics in an infinite population, for each of the four social norms we have studied. One insight this immediately provides is that whether or not selection favors adherence is determined by the size of the benefit-to-cost ratio $b/c$ compared to a critical value $\rho$ that depends on the error rates, social norm, and the institution's size and strictness (Methods Eq. (15)).

Figure 5 shows the growth rate of a rare adherent type (DISC-ADHERE) in a population otherwise comprised of private assessors (DISC-PRIVATE). The benefit-to-cost ratio and the error rates can have substantial effects on the magnitude of selection for or against institutional adherents. Generally speaking, for both strict and tolerant institutions, variation in $b/c$ tends to have larger effects on adherent growth compared to variation in errors of assessment ($e_2$) or errors of strategy implementation ($e_1$). These analyses across a wide range of parameters uncover several new qualitative phenomena – conditions where the benefit-to-cost ratio determines whether adherence is selectively favored or disfavored. Under tolerant Stern Judging, we have already seen that selection disfavors institutional adherents for $b/c = 5$ with our default error rates (Fig. 4), in agreement with replicator-dynamic predictions (Fig. 5); and yet replicator dynamics imply adherence to tolerant Stern Judging will be selectively favored for sufficiently large benefit-to-cost ratios (e.g., $b/c > 50$, Fig. 5). We verified this prediction by Monte Carlo simulations in finite populations (Supplementary Fig. 8). Likewise, under strict Shunning, replicator dynamics imply that selection will favor institutional adherence when $b/c < 50$, but selection will disfavor adherence

for $b/c > 50$ (Fig. 5), and under tolerant Shunning, replicator dynamics imply selection will favor adherence more strongly when $b/c$ is larger (Fig. 5). All of these analytical predictions from replicator dynamics were verified by Monte Carlo simulations in finite populations (Fig. 4 and Supplementary Fig. 8). Similar results hold for an "internal board" of institution members (Supplementary Figs. 7, 9, and 13).

## Discussion

Much of human prosocial behavior is guided by social norms and informal moral codes of fairness[31,41,42]. Societies also adopt formal institutions that reinforce these informal constructs to enforce conditional cooperation and resolve conflict. For instance, local institutions help resolve public goods dilemmas in irrigation systems[43] and management of forest commons[44]. Likewise, e-commerce companies aggregate individual assessments of the reputations of buyers and sellers, providing a public broadcast to a large community of users[12]. Credit bureaus synthesize and publicize the reputations of borrowers, so that lending agencies can choose to reward cooperative behavior with easy access to future capital[45]. Analogous institutions even existed in the era of pre-state trade[13,14].

Adherence to public information about reputations is the default assumption for the theory of cooperation by indirect reciprocity, in which cooperation is conditioned on the reputations of potential exchange partners. In practice, however, reputations could be assigned by independent, private monitoring of social interactions, and so they may be observer-dependent. When individuals do not adhere to a public broadcast and choose private monitoring instead, this causes the breakdown of

cooperation[25–27]. Thus, understanding how institutions of judgment affect rates of cooperation, and when individuals will choose to trust an institutional broadcast, is a required component for a theory of cooperation by indirect reciprocity.

We have analyzed how the size and strictness of an institution of public monitoring affect the level of cooperation in an adherent society. But the question remains: why would people choose to trust moral judgments provided by the institutional observers, rather than trusting their own moral compass? To address this question we have also studied whether adherence to a public institution will spread by social contagion in a population that initially relies on private judgments. Our key finding is that adherence to a centralized institution of assessment can often spread when rare, as individuals that abandon their private beliefs are on average more successful than their peers. If the entire population trusts the public monitoring system then the rate of disagreement about the moral standing of individuals is reduced, which eliminates unjustified punishment. Nonetheless, the specific form of public institution that provides the greatest benefit depends on the social norm. Larger institutions tend to outperform small ones, and they can perform far better than broadcasting the judgments of a single individual. Tolerant institutions stimulate more cooperation for certain social norms of moral judgment, whereas strict institutions perform better for other norms.

Our results provide a theoretical justification for a key assumption in the theory of indirect reciprocity: adherence to public monitoring. We also find that we can specify the size and strictness of an institution so as to produce high levels of cooperation, even for social norms that previous studies have found to perform poorly, such as Scoring, Shunning, and Simple Standing. The optimal size and strictness of the institution depend on the prevailing social norm. For example, societies that adhere to Scoring or Shunning norms can achieve rates of cooperation as high as those found in societies governed by Stern Judging, provided the centralized monitoring system consists of more than a single individual and is highly tolerant.

A different solution to the problem of private monitoring, and consequent breakdown in cooperation, is empathetic perspective taking[27]. But it is difficult to know how effective empathy is at promoting cooperation in the face of cognitive costs and the potential for cheating or manipulation. Nonetheless, once individuals have acquired a capacity for empathy, this internal mechanism often "crowds out" selection that would otherwise favor adherence to an public institution of centralized monitoring (Fig. 4). The extent of interference between these internal and external mechanisms of moral assessment, as well as the precise relationship to motivation crowding in the economics literature, remains a topic for future research.

Our analysis relies on mathematical models and computer simulations to describe the dynamics of reputations and behavior. Although the theory of indirect reciprocity is enshrined in such models[34], they are a crude approximation of the complexities of real-life behavior and the nuances of moral judgment. At best, these models aim to retain the key qualitative features of social dilemmas. Predictions from models with this level of granularity have been tested in several notable experiments[7,46]. And yet, before any practical recommendations can be made, further controlled laboratory experiments and field studies on reputations and behavior (e.g.,[24,45]) are required to bring theories of cooperation into contact with empirical observation.

Our study does not address how an institution of public monitoring is established to begin with. Instead, we have focused on whether, and when, adherence to an existing public broadcast is beneficial. The question of establishing an institution de novo is perhaps more difficult – and more properly a question in political science than in evolutionary game theory. In practice, modern societies that possess formal systems of governance typically form public institutions by legislation and taxation. Although the problem of forming an institution is outside the scope of our study, we can nonetheless provide some rudimentary insights by quantifying the level of taxation that rational individuals would willingly pay to support such an institution, in order to reap the rewards of cooperation it provides. A defecting population of private assessors (who obtain payoff zero) may be selectively replaced by a cooperative population of institutional adherents (who obtain payoff $b - c$) provided the benefit-to-cost ratio $b/c$ exceeds a critical value $\rho$ (Methods Eq. (15)). Each individual in the population would therefore be willing to pay a tax as large as $T = b - c\rho$ per round to support the operation of such a public institution.

Our analysis also neglects the problem of institutional corruption. For example, what is to stop a member of the institution from engaging in side deals with individuals who seek to extract a good reputation broadcast in exchange for a bribe? Here again we can at least make a rudimentary calculation for suppressing corruption by withholding tax payment to institution members. As described above, each member of the institution can expect to receive an income of $NT/Q$ per round, in a society of $N$ rational individuals who will willingly pay a tax to support an institution of public monitoring. (Each of $N$ rational individuals will willingly contribute tax $T$ per round to fund a public institution that engenders stable mutual cooperation instead of defection. The gross proceeds of this tax are shared equally among the $Q$ members of the institution.) And so, even in the simple one-shot setting, if corruption carries a chance of discovery $p$ that is penalized by loss of income, then an institution member will be robust against a bribe $B$ provided $(1 - p)B < NT/Q$. This simple calculus for suppressing corruption relies on an institution supported by taxation, achieved by mutual agreement out of rational self-interest. In this setting, punishment of a corrupt member (by depriving her payment, and thus reducing the public tax burden) is not costly but actually advantageous to the public. This approach avoids the second-order free rider problem of costly punishment[47,48], which would be required to explain the bottom-up formation of an honest institution in societies lacking structures for governance – an important topic for future research.

Although the theory of indirect reciprocity is typically limited to pairwise interactions, the simple idea of behavior conditioned on partner reputation is a compelling explanation for cooperation. The role of reputations in $n$-player interactions, such as in public goods games, remains less explored than in pairwise interactions (but see[48]). Likewise, the concept of a social norm in this context – that is, the moral heuristic that determines which types of actions result in assigning someone a good or bad reputation – requires further theoretical and empirical study. Very little is known about the origin and spread of such norms. The standard assumption in the literature is that norms are universally accepted or imposed externally and do not change over time. There is ongoing discovery of new, effective social norms[35], and several studies of norm competition[49,50]. These studies have begun to address how the definition of moral good may itself evolve via random exploration and social contagion. But it remains unclear which social norms of moral judgment will emerge when norms evolve from personal beliefs in societies, either with or without public institutions of assessment. We also lack a mechanistic theory for how the definitions of moral codes are transmitted among individuals, or whether any a priori restrictions limit the norms that individuals can choose or transmit. These questions await future research.

## Methods

**1. Strategy evolution under public monitoring.** We start by analyzing the evolutionary dynamics of behavioral strategies in an infinite population, assuming complete adherence to a public institution of moral assessment. We use classical replicator dynamics[32,33] to describe how the frequencies of three strategic types change in time. We consider the following strategies: always cooperate (ALLC, denoted $X$), always defect (ALLD, denoted $Y$), and discriminate (DISC, denoted $Z$). Unlike ALLC or ALLD, the DISC type conditions behavior on reputations: DISC cooperates with individuals that have a good reputation and defects with individuals that have a bad reputation.

*1.1 Replicator dynamics.* The frequencies of the three strategic types, $f_X$, $f_Y$, and $f_Z$ change according to their fitness relative to the mean population fitness:

$$\dot{f}_i = f_i(\Pi_i - \bar{\Pi}), \text{ with}$$
$$\bar{\Pi} = \sum_{j \in \{X,Y,Z\}} f_j \Pi_j \quad (1)$$

where $i \in \{X, Y, Z\}$ denotes the $i$th strategic type and $\Pi_i$ denotes the mean fitness of type $i$.

The fitness of each individual is determined by interactions in the donation game with all other individuals in the population. Each individual interacts once as donor and once as recipient with all other individuals in the population. Each individual derives a benefit $b$ from interaction with every cooperator and from every interaction with a discriminator who views them as good (i.e., if the focal individual has a good institutional reputation). Cooperators always pay the cost $c$ of cooperation, and discriminators pay the cost only to individuals who have a good reputation; defectors never pay the cost. Finally, with probability $e_1$, an individual who intends to cooperate will accidentally defect. As a result, the mean fitness of individuals with each strategic type is given by

$$\Pi_X = (1 - e_1)b(f_X + f_Z G_X) - (1 - e_1)c$$
$$\Pi_Y = (1 - e_1)b(f_X + f_Z G_Y) \quad (2)$$
$$\Pi_Z = (1 - e_1)b(f_X + f_Z G_Z) - (1 - e_1)Gc,$$

where $G_i$ denotes the fraction of individuals of type $i$ who have good institutional reputations and $G = \sum_i f_i G_i$ denotes the total fraction of the population with a good institutional reputation.

*1.2 Reputation dynamics.* We assume that reputation frequencies reach equilibrium before strategy frequencies change – that is, we assume the timescale of reputation updating is fast compared to that of strategic updates[51]. In this section we describe the equilibrium reputations that emerge, assuming complete adherence to the public institution of moral assessment.

Reputations in our model are publicly known and determined by the consensus view by provided by an "institution". The institution consists of $Q$ members. Each round, each institution member observes a independent, random interaction of each individual acting as a donor. As in[51], we assume that each observer makes her moral evaluation of a donor's action based on an independently observed social interaction.

An institution member updates the reputation of a donor according to the donor's action (cooperate or defect) towards a recipient. The resulting reputation that member ascribes to the donor is governed by the social norm of the population. After forming their individual assessments, if at least a fraction $q$ of the institution members agree that an individual's reputation is good, then the individual's reputation is broadcast as good to the entire population. Otherwise, it is broadcast as bad. Table 1 summarizes the probability that an institutional observer will assess a donor's reputation as good, depending on the observer's view of the recipient, the donor's intended action, and the social norm. These probabilities account for the two types of errors that occur in our model: a donor's cooperative intention is erroneously executed as defection with probability $e_1$, and

an observer erroneously assigns a bad reputation instead of a good reputation, and vice versa, with probability $e_2$.

The probabilities in Table 1 will help us to derive the equilibrium proportion of individuals of type $i$ that are viewed as having a good reputation in the eyes of an arbitrary observer, which we denote $g_i$. The views of the $Q$ observers that comprise the institution will then collectively determine the institutional broadcasts, and thus $G_i$, as described below.

The equilibrium proportion of type $i$ that is assigned a good reputation by an arbitrary observer can be expressed as the solution to a system of three simultaneous equations, one for each $i \in \{X, Y, Z\}$. We derive these equations in detail for the Stern Judging norm, referring to Table 1 as needed. (Analogous derivations apply for the three other norms.) Consider first the chance, $g_X$, that an unconditional cooperator will be assigned a good reputation by a observer. An unconditional cooperator ends up with a good reputation in the following two ways:

1. She interacts with someone who has a good reputation (probability $G$), intending to cooperate, and is successfully assigned a good reputation by the observer (probability $\epsilon$, from Table 1).
2. She interacts with someone who has a bad reputation (probability $1 - G$), intending to cooperate, and is erroneously assigned a good reputation (probability $1 - \epsilon$, from Table 1).

Likewise, an unconditional defector ends up with a good reputation in two ways:

1. She interacts with someone who has a good reputation (probability $G$), intending to defect, and is erroneously assigned a good reputation (probability $e_2$, from Table 1).
2. She interacts with someone who has a bad reputation (probability $1 - G$), intending to defect, and she is correctly assigned a good reputation (probability $1 - e_2$, from Table 1).

Finally, a discriminator ends up with a good reputation in two ways:

1. She interacts with someone who has a good reputation (probability $G$), intending to cooperate, and is correctly assigned a good reputation (probability $\epsilon$, from Table 1).
2. She interacts with someone who has a bad reputation (probability $1 - G$), intending to defect, and is correctly assigned a good reputation (probability $1 - e_2$, from Table 1).

In summary, for a population following the Stern Judging norm, the equilibrium frequencies of good individuals in each strategic type from the perspective of an arbitrary observer satisfy the following equations:

$$g_X = \epsilon G + (1 - \epsilon)(1 - G)$$
$$g_Y = e_2 G + (1 - e_2)(1 - G) \quad (3)$$
$$g_Z = \epsilon G + (1 - e_2)(1 - G).$$

An analogous derivation for Simple Standing yields

$$g_X = \epsilon G + (1 - e_2)(1 - G)$$
$$g_Y = e_2 G + (1 - e_2)(1 - G) \quad (4)$$
$$g_Z = \epsilon G + (1 - e_2)(1 - G).$$

Likewise, for Scoring we have

$$g_X = \epsilon$$
$$g_Y = e_2 \quad (5)$$
$$g_Z = \epsilon G + (1 - e_2)(1 - G).$$

Finally, for Shunning we have

$$g_X = \epsilon G + e_2(1 - G)$$
$$g_Y = e_2 \quad (6)$$
$$g_Z = \epsilon G + e_2(1 - G).$$

These equations correspond to the expressions under private assessment with complete empathy from prior studies[27], with the exception that $g = \sum_i f_i g_i$, the proportion of the population with a good reputation in the eyes of an arbitrary observer, is replaced here by $G = \sum_i f_i G_i$, the proportion of the population with a good institutional reputation.

For each social norm, the system of equations above is closed once we specify how $G$ depends on $g_X$, $g_Y$, $g_Z$ and on the size and strictness of the institution. We consider two limiting cases: very strict institutions that broadcast an individual as good only if all members agree she is good ($q > (Q - 1)/Q$) and very tolerant institutions that broadcast an individual as good provided at least one member views her as good ($q < 1/Q$). In these two respective cases we have

$$G_i = \begin{cases} 1 - (1 - g_i)^Q & \text{for tolerant institutions}, i \in \{X, Y, Z\}, \\ g_i^Q & \text{for strict institutions}, i \in \{X, Y, Z\}. \end{cases} \quad (7)$$

**Table 1 Probability $P_k$ that an observer will assign a donor a good reputation, under each of four social norms $k \in$ [stern judging, simple standing, scoring, shunning].**

| Observer view of recipient | Donor intent | $P_{SJ}$ | $P_{SS}$ | $P_{SC}$ | $P_{SH}$ |
|---|---|---|---|---|---|
| Good | Cooperate | $\epsilon$ | $\epsilon$ | $\epsilon$ | $\epsilon$ |
| Good | Defect | $e_2$ | $e_2$ | $e_2$ | $e_2$ |
| Bad | Cooperate | $1 - \epsilon$ | $1 - e_2$ | $\epsilon$ | $e_2$ |
| Bad | Defect | $1 - e_2$ | $1 - e_2$ | $e_2$ | $e_2$ |

Here, $\epsilon = (1 - e_2)(1 - e_1) + e_2 e_1$ is the probability that an individual who intends to cooperate with a recipient who has a good reputation is ultimately themselves assigned a good reputation – either because they successfully cooperate and are correctly assigned a good reputation (first term) or accidentally defect and are erroneously assigned a good reputation (second term).

For completeness' sake, we provide the expression for arbitrary $q$ and $Q$:

$$G_i = \sum_{k=\text{ceil}(qQ)}^{Q} \binom{Q}{k} g_i^k (1-g_i)^{Q-k}. \tag{8}$$

The resulting system of equations for $g_X, g_Y$, and $g_Z$ can be solved by radicals when $Q \leq 2$. More generally, a unique feasible solution exists for any $Q$, and it can be computed numerically by iterating the above system of equations after choosing any initial value $(g_X^0, g_Y^0, g_Z^0) \in (0,1)^3$. Successive iterates $(g_X^k, g_Y^k, g_Z^k)$ remain in $[0,1]^3$ and form a contraction: the equations for $g_X, g_Y, g_Z$ are each convex combinations of elements in $(0,1)$, provided $e_2 > 0$ and $e_1 > 0$. Solving this system determines the equilibrium frequency of good individuals of each strategic type from the eyes of an arbitrary observer ($g_i$) and therefore also yields the institutional broadcast ($G_i$, from Eq. (7), and $G = \sum_i f_i G_i$). Substituting these expressions into the replicator equation (Eqs. (1) and (2)) provides the dynamics of strategy frequencies $f_i$, allowing us to compute selection gradients, strategic equilibria, and basins of attraction, as shown in Fig. 2.

## 2. Competition between institutional adherents and non-adherents.
Whereas the analysis in Section "Strategy evolution under public monitoring" assumes complete adherence to the reputations broadcast by the public institution, in this section we now accommodate a mixture of individuals, of which some condition their behavior on their own private assessment of recipient reputations and others condition their behavior on the institutional broadcast. This distinction has no effect on the behavior of unconditional cooperators or defectors (ALLC or ALLD), but it does affect the behavior of discriminators. And so, henceforth we divide the discriminators into two types: *adherents* (DISC-ADHERE, denoted $Z_a$) and *non-adherents* (DISC-PRIVATE, denoted $Z_n$).

Adherents use the reputations broadcast by the institution when determining whether to cooperate or defect with an individual. Non-adherents do not: they use their private view of the individual, which was formed by observing a random interaction in which that individual acted as a donor. Here we develop replicator dynamics to describe competition between these two types of discriminators, adherents $Z_a$ and non-adherents $Z_n$. (Eventually we extend this to describe competition between all four types, $X$, $Y$, $Z_a$, and $Z_n$.)

We consider two alternative assessment protocols for the institution members themselves. (This distinction was not relevant in the context of complete adherence, in the preceding section.) In particular, we consider two possibilities for the institution:

1. The institution may be comprised of an *internal board* of $Q$ individuals sampled from the population. In this case the internal board may consist of adherents, non-adherents, or a mix thereof. When a board member is an adherent, they will use the institutional assessment of a recipient when updating their opinion of a donor's reputation. When a board member is a non-adherent, they will use their own private view of a recipient, which may differ from the public institutional one, when assessing a donor.

2. The institution may alternatively be comprised of an *external board* of $Q$ individuals who are all adherents, irrespective of the strategy composition of the population. External board members rely solely on the institutional public broadcast of the recipient when formulating their opinion of a donor's reputation. We describe this configuration as "external" because the institution members act "outside" of the population, using institutional assessments to determine reputations irrespective of the actual composition of the population. Equivalently, the external board can be interpreted as an institution whose members are inside the population itself, provided the institution members diligently compartmentalize their own personal reputation assessments (when acting as a donor) from the institutional assessment (when they contribute to updating the institution's view of a donor).

*2.1 Replicator dynamics.* Because adherents and non-adherents may not cooperate and defect in the same situations, their reputation dynamics must be considered separately. Let $g_{i,j}$ denote the proportion of individuals following strategy $i$ that are seen as having a good reputation by an individual with strategy $j$. As before, we let $G_i$ denote the proportion of the individuals of type $i$ that have a good institutional reputation. Define

$$
\begin{aligned}
G &= \sum_i f_i G_i, \\
g_\bullet &= \sum_i f_i g_{i,Z_n}, \\
\gamma_2 &= \sum_i f_i G_i g_{i,Z_n}, \\
g_2 &= \sum_i f_i g_{i,Z_n}^2.
\end{aligned} \tag{9}
$$

Here, $G$ is (as previously) the proportion of the total population with a good institutional reputation, $g_\bullet$ is the proportion of the total population that a non-adherent sees as having a good reputation, $\gamma_2$ is the probability that an institutional adherent and non-adherent agree that a given individual's reputation is good, and $g_2$ is the probability that two non-adherents agree that a given individual's

reputation is good. The strategic types $i$ in Eq. (9) range over $Z_a$ and $Z_n$ in this section (but they will additionally include $X$ and $Y$ in Section "Multiple strategic types").

The mean fitness of individuals of type $Z_n$ and $Z_a$ can now be written as

$$
\begin{aligned}
\Pi_{Z_n} &= (1-e_1) b (f_{Z_n} g_{Z_n,Z_n} + f_{Z_a} G_{Z_n}) - c(1-e_1) g_\bullet \\
\Pi_{Z_a} &= (1-e_1) b (f_{Z_n} g_{Z_a,Z_n} + f_{Z_a} G_{Z_a}) - c(1-e_1) G,
\end{aligned} \tag{10}
$$

which notably differ from the expressions in the context of complete adherence. Finally, the replicator dynamics are given by

$$
\begin{aligned}
\dot{f}_i &= f_i(\Pi_i - \bar{\Pi}), \text{ with} \\
\bar{\Pi} &= \sum_{j \in \{Z_n, Z_a\}} f_j \Pi_j.
\end{aligned} \tag{11}
$$

Note that $g_{Z_n,Z_a}$ and $g_{Z_a,Z_a}$ do not appear in the fitness expressions or replicator dynamic equations. This is because adherents' private assessments of reputations do not directly determine their behavior: their behavior is governed solely by institutional assessments.

*2.2 Reputation dynamics.* Reputation dynamics of a population containing both adherents ($Z_a$) and non-adherents ($Z_n$) must account for all possible configurations of an observer assessing a donor interacting with a recipient (Table 2). By cross-referencing Table 2 with Table 1, we obtain expressions for $g_{i,j}$, the proportion of individuals following strategy $i$ that are seen as good by an individual of strategy $j$. For example, under the Stern Judging norm, we have the following system of equations for equilibrium values of $g_{i,j}$:

$$
\begin{aligned}
g_{Z_a,Z_a} &= G\epsilon + (1-G)(1-e_2), \\
g_{Z_n,Z_n} &= \gamma_2 \epsilon + (g_\bullet - \gamma_2) e_2 + (G - \gamma_2)(1-\epsilon) + (1 - g_\bullet - G + \gamma_2)(1-e_2), \\
g_{Z_a,Z_n} &= \gamma_2 \epsilon + (g_\bullet - \gamma_2)(1-\epsilon) + (G - \gamma_2) e_2 + (1 - g_\bullet - G + \gamma_2)(1-e_2), \\
g_{Z_n,Z_n} &= g_2 \epsilon + (g_\bullet - g_2)(e_2 + 1 - \epsilon) + (1 - 2g_\bullet + g_2)(1-e_2).
\end{aligned} \tag{12}
$$

Note that for a population consisting entirely of non-adherents, these equations reduce to the case of non-empathetic private assessment that has been studied previously[27]. Analogous equations for Simple Standing, Shunning, and Scoring can be immediately obtained in the same manner.

The system of equations for equilibrium $g_{i,j}$ is closed once we specify how individual views are aggregated to produces institutional assessments, $G_i$. In the case of an external board whose members always make assessments based on the institutional view of a recipient, we have the following:

$$
G_i = \begin{cases} 1 - (1 - g_{i,Z_a})^Q & \text{for tolerant institutions, external board,} \\ g_{i,Z_a}^Q & \text{for strict institutions, external board.} \end{cases} \tag{13}
$$

For the case of an internal board, the composition of adherents and non-adherents among institution members is determined by randomly drawing from the general population. And so in this case we have

$$
G_i = \begin{cases} \sum_{k=0}^{Q} \binom{Q}{k} f_{Z_a}^{Q-k} f_{Z_n}^k [1 - (1 - g_{i,Z_a})^{Q-k} (1 - g_{i,Z_n})^k] & \text{for tolerant institutions, internal board,} \\ \sum_{k=0}^{Q} \binom{Q}{k} f_{Z_a}^{Q-k} f_{Z_n}^k g_{i,Z_a}^{Q-k} g_{i,Z_n}^k & \text{for strict institutions, internal board.} \end{cases} \tag{14}
$$

In either case, we can substitute the expressions for $G_i$ and $G = \sum_i f_i G_i$ into Eq. (12) to obtain a closed system of equations for the equilibrium values of $g_{ij}$. For any $Q$, this system of equations has a unique solution in $[0,1]^4$, which can be constructed by numerical iteration, because the right-hand side of each equation is a convex combination of elements in $(0,1)$, provided $e_2 > 0$ and $e_1 > 0$. The equilibrium reputations can then be substituted into the replicator equation (Eq. (11)) to describe the dynamics of strategy frequencies $f_i$ over time.

Note that the condition for selection to favor (positive) growth of adherents, $Z_a$, is given by

$$
\begin{aligned}
0 &< \Pi_{Z_a} - \Pi_{Z_n} \\
\therefore 0 &< b(f_{Z_n}[g_{Z_a,Z_n} - g_{Z_n,Z_n}] + f_{Z_a}[G_{Z_a} - G_{Z_n}]) - c(G - g_\bullet)
\end{aligned}
$$

And so for $b$ ($f_{Z_n}[g_{Z_a,Z_n} - g_{Z_n,Z_n}] + f_{Z_a}[G_{Z_a} - G_{Z_n}] > 0$, the condition for positive growth of $Z_a$ is

$$
\frac{b}{c} > \frac{G - g_\bullet}{f_{Z_n}[g_{Z_a,Z_n} - g_{Z_n,Z_n}] + f_{Z_a}[G_{Z_a} - G_{Z_n}]}; \tag{15}
$$

otherwise, it is

$$
\frac{b}{c} < \frac{G - g_\bullet}{f_{Z_n}[g_{Z_a,Z_n} - g_{Z_n,Z_n}] + f_{Z_a}[G_{Z_a} - G_{Z_n}]}.
$$

The only parameters that appear directly in these conditions are the game payoffs $b$ and $c$, whereas $e_2$ and $e_1$ are involved in the expressions for reputations on the right-hand side. According to these inequalities, the question of whether selection favors adherence or not depends on the benefit-to-cost ratio $b/c$ compared to a critical value $\rho$ that depends on the error rates, the social norm, and the institution's size and strictness. These expressions determine whether adherents

**Table 2 Possible configurations of an observer, donor, and recipient.**

| Observer strategy | Observer view of recipient | Donor strategy | Donor view of recipient | Probability of configuration | Donor intent |
|---|---|---|---|---|---|
| $Z_a$ | Good | $Z_a$ | Good | $G$ | Cooperate |
| $Z_a$ | Good | $Z_a$ | Bad | $0$ | Defect |
| $Z_a$ | Bad | $Z_a$ | Good | $0$ | Cooperate |
| $Z_a$ | Bad | $Z_a$ | Bad | $1-G$ | Defect |
| $Z_a$ | Good | $Z_n$ | Good | $\gamma_2$ | Cooperate |
| $Z_a$ | Good | $Z_n$ | Bad | $G-\gamma_2$ | Defect |
| $Z_a$ | Bad | $Z_n$ | Good | $g_\bullet - \gamma_2$ | Cooperate |
| $Z_a$ | Bad | $Z_n$ | Bad | $1-G-g_\bullet+\gamma_2$ | Defect |
| $Z_n$ | Good | $Z_a$ | Good | $\gamma_2$ | Cooperate |
| $Z_n$ | Good | $Z_a$ | Bad | $g_\bullet - \gamma_2$ | Defect |
| $Z_n$ | Bad | $Z_a$ | Good | $G-\gamma_2$ | Cooperate |
| $Z_n$ | Bad | $Z_a$ | Bad | $1-G-g_\bullet+\gamma_2$ | Defect |
| $Z_n$ | Good | $Z_n$ | Good | $g_2$ | Cooperate |
| $Z_n$ | Good | $Z_n$ | Bad | $g_\bullet - g_2$ | Defect |
| $Z_n$ | Bad | $Z_n$ | Good | $g_\bullet - g_2$ | Cooperate |
| $Z_n$ | Bad | $Z_n$ | Bad | $1-2g_\bullet+g_2$ | Defect |

The fifth column indicates the probability of each configuration, and the sixth column indicates the donor's intent (which follows directly from the donor's view of the recipient). Cross-referencing with Table 1 yields the probability that the donor will be assessed as good in the eyes of the observer.

will invade, and analogous expressions determine whether the equilibrium $Z_a = 1$ is stable.

### 2.3 Multiple strategic types.
Finally, we can extend this analysis to include cooperators, defectors, and both flavors of discriminators. As before, we differentiate between the reputation assessments of institution adherents and non-adherents. Under the Stern Judging norm, the reputations of cooperators and defectors will be given by

$$
\begin{aligned}
g_{X,Z_a} &= G\epsilon + (1-G)(1-\epsilon),\\
g_{X,Z_n} &= g_\bullet\epsilon + (1-g_\bullet)(1-\epsilon),\\
g_{Y,Z_a} &= Ge_2 + (1-G)(1-e_2),\\
g_{Y,Z_n} &= g_\bullet\epsilon + (1-g_\bullet)(1-e_2).
\end{aligned}
\tag{16}
$$

Reputations under other norms can be derived analogously. The replicator dynamics can be extended from Eq. (10):

$$
\begin{aligned}
\Pi_X &= (1-e_1)b(f_X + f_{Z_n}g_{X,Z_n} + f_{Z_a}G_X) - c(1-e_1)\\
\Pi_Y &= (1-e_1)b(f_X + f_{Z_n}g_{Y,Z_n} + f_{Z_a}G_Y)\\
\Pi_{Z_n} &= (1-e_1)b(f_X + f_{Z_n}g_{Z_n,Z_n} + f_{Z_a}G_{Z_n}) - c(1-e_1)g_\bullet\\
\Pi_{Z_a} &= (1-e_1)b(f_X + f_{Z_n}g_{Z_a,Z_n} + f_{Z_a}G_{Z_a}) - c(1-e_1)G.
\end{aligned}
\tag{17}
$$

### 3. Monte Carlo simulations.
To study cooperation in finite populations of institution adherents, we performed a series of Monte Carlo simulations implemented in Julia. Each population consists of $N = 50$ individuals, all following the same social norm but each with its own strategy: ALLC, ALLD, or DISC. In each successive discrete generation, every individual interacts with every other individual in three roles: once as a donor, once as a recipient, and once as an observer.

First, each individual plays a single round of the donation game with every member of the population, taking actions according to their own strategy and the (public) reputation of the recipient. We include self-interactions, although individuals cannot derive any net benefit from self-interactions. Individuals who intend to cooperate will accidentally defect with probability $e_1$. Each individual pays a cost $c$ for every game in which they cooperate and accrues a benefit $b$ from every game in which their interaction partner cooperates.

The institution of public monitoring consists of $1 \le Q \le N$ designated players, each with their own subjective view (either "good" or "bad") of the reputations of the rest of the population. The institution members remain the same throughout a given simulation. For every member $i$ of the population, each institution member observes a single random interaction in which $i$ acted as a donor. They then update their opinion of $i$ based on $i$'s action, the public reputation of the recipient $j$, and the social norm. With probability $e_2$ they assign the wrong reputation. If more than a threshold number $qQ$ of institution members consider an individual to be "good", that individual's reputation is publicly broadcast as "good". Otherwise, it is publicly broadcast as "bad". These updated public reputations are used by the entire population in the following round to determine their behavior.

Selection and drift are implemented as a social contagion process using pairwise comparison. After the reputation updating step, a random pair of individuals $i$ and $j$ is chosen. Player $i$ adopts the strategy of $j$ with probability $1/(1 + \exp(-w[\Pi_j - \Pi_i]))$, where $w$ is a selection strength parameter and $\Pi_i$ and $\Pi_j$ are the average payoffs earned by $i$ and $j$ (that is, the total payoff divided by $N$ interactions). We set $w = 1$ for

all simulations. Finally, with probability $\mu = 0.025$, an individual is randomly chosen to switch (or "mutate") to a random strategy, following[20].

We initialize each simulation with random strategies and random reputations. We run each simulation for 10,000 generations and record the mean cooperation rate over the last 5000 generations, then average it over 2500 independent replicate simulations. We compared our results to simulations in which individuals rely exclusively on their own private reputation assessments but may, with probability $E$, empathetically adopt the perspective of the donor in each interaction, as described by Radzvilavicius et al.[27]. The results of these simulations were used to produce Fig. 3.

Next, to study the evolution of adherence to a public institution, we adjusted the simulation to account for a mix of adherents and non-adherents. We initialized each population to consist solely of discriminators but added a binary trait corresponding to whether an individual uses their own private reputation assessments (non-adherent, $Z_n$) or the institutionally broadcast public assessments (adherent, $Z_a$). (Mutation is set to zero in these simulations.) Each population initially consists entirely of non-adherents: after an equilibration period (100 generations), a single adherent is introduced in the population. In each subsequent generation, a random pair of individuals $i$ and $j$ is selected, and $i$ adopts $j$'s type (either $Z_a$ or $Z_n$) with probability $1/(1 + \exp(-w[\Pi_j - \Pi_i]))$, as above. In addition, we either mandate that all $Q$ institution members be adherents (external board) or allow them to use the assessment strategy of the first $Q$ individuals in the population (internal board). Finally, we permitted empathetic evaluation with probability $E$ as outlined in[27]. These simulations continued until the institution following trait either fixes or goes extinct. We record the proportion of fixations observed among 2500 replicate simulations per parameter combination. These simulations were used to produce Fig. 4 and Supplementary Figs. 7, 8, 9, and 13.

To measure the stability of institutional adherence in finite populations, we repeated the previous set of simulations, except that we initialized the entire population as institutional adherents ($Z_a$) and introduced a single non-adherent ($Z_n$). For each parameter combination, we initialized 2500 such simulations and allowed each one to run until the adherent type either fixed or went extinct, then recorded the proportion of simulations in which the adherent fixed. These simulations were used to produce Supplementary Fig. 14.

To further probe the stability of institutional adherence in finite populations, we repeated this set of simulations, except that, after reputation equilibration, a random number $\in [1, N - 1]$ of individuals was introduced with institution-adherence (type $Z_a$). A total of 2500 replicate simulations was performed for each parameter combination and permitted to run until the adherence trait was driven to either fixation or extinction. We recorded the mean fixation frequency over these replicate simulations. These simulations were used to produce Supplementary Figs. 1 and 2.

Finally, we conducted simulations to estimate the fixation probability of adherents ($Z_a$) introduced into a population containing a mixture of non-adherent discriminators ($Z_n$), unconditional cooperators ($X$), and unconditional defectors ($Y$). The protocol for these simulations was identical to that described in the paragraph above, except that ALLC and ALLD players do not condition their behavior on reputations. We estimated fixation probability starting from four different configurations, corresponding to different positions on the $X, Y, Z_a$ simplex: simulations in which the initial frequencies were set to 16/50, 17/50, and 17/50 respectively (roughly the center of the simplex), and three sets of simulations, each starting with one strategy frequency set to 46/50 and the other two set to 2/50 (near the corners of the simplex). Reputations were permitted to equilibrate for 100

generations without strategy updating. A single DISC-PRIVATE individual $Z_n$ was then switched to become DISC-ADHERE, and the simulation was permitted to run until DISC-ADHERE either fixed or went extinct. A total of 2500 replicate simulations were used to estimate the fixation probability of DISC-ADHERE under these conditions. These simulations were used to produce Supplementary Figs. 3, 4, 5, and 6.

**Reporting summary**. Further information on research design is available in the Nature Research Reporting Summary linked to this article.

## Data availability
There are no empirical data associated with this study.

## Code availability
All computer codes for simulations described in this study are freely available at the following public repository: https://github.com/tkessinger/institutions.

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

## Acknowledgements
We thank three anonymous referees for their insights and constructive comments.

## Author contributions
Designed the study, executed the analyses, and wrote the paper: A.L.R., T.A.K., J.B.P.

## Competing interests
The authors declare no competing interests.

## Additional information

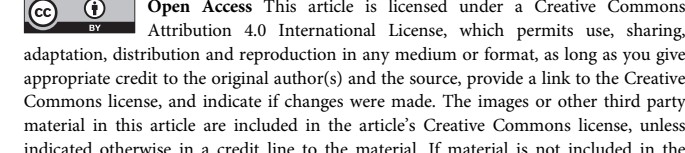

