## [Peer Review File · Nature Communications]

REVIEWER COMMENTS

Reviewer #1 (Remarks to the Author):

Summary.

The paper explores the impact of institutions on the evolution of indirect reciprocity. Indirect reciprocity is one of the major mechanisms to explain why humans may sometimes help complete strangers. The idea is that by helping someone, people increase their reputation. A good reputation, in turn, makes it more likely to receive cooperation in future.

A serious limitation of indirect reciprocity, however, seems to be that it makes rather stringent assumptions on how individuals share information. In the classical model of Ohtsuki and Iwasa it was shown that cooperation can evolve if all information is disseminated by a central observer. One consequence of this assumption is that all individuals agree on which reputation should be assigned to any given population member. In contrast, in a few recent studies it was shown that cooperation has problems to evolve if population members form their opinions independently. In that case, slight disagreements between players can quickly proliferate. In the long run, these disagreements can lead to the breakdown of cooperation.

In this present study, the authors connect these two extremes of public and private information. Using a mixture of analytical methods and simulations, the paper explores how public information sharing might evolve in the first place. Once such central institutions are in place, cooperation can evolve more easily than in a completely private setup. In fact, under some circumstances cooperation can prevail even for social norms that have been previously thought to be insufficient for the evolution of stable cooperation (Scoring and Shunning).

General evaluation.

The paper makes a very interesting contribution to the field. It provides a possibility to reconcile the findings of Ohtsuki and Iwasa with more recent work on private information. The used methods are sound and the main text is a pleasure to read.

Having said that, there are also a few places where I think the manuscript can be improved. For example, while the model depends on quite a few parameters, only two parameters (q and Q) are systematically varied throughout the study. This makes it somewhat difficult to assess how general and robust the shown results are. In addition, I believe the discussion section would benefit from a more critical evaluation of some of the models' assumptions. Finally, I would like to encourage the authors to provide more detailed information in their methods section.

Once these points I addressed, I think the paper makes a very welcome contribution to the field. Such a paper is certainly worth being published in Nature Communications. Please find more details below.

Major suggestions.

(-) Robustness: The model depends on a number of parameters. For example, already the simple baseline model depends on the benefit of cooperation, and on different error rates. The stochastic model additionally depends on the mutation rate, the selection strength, and population size. I would appreciate to see some indication on whether the presented results are robust with respect to changes in these parameters. Also, what is the qualitative effect of these parameters? For example, is the outcome the same if assessment errors occur at a much higher rate than implementation errors?

In general I was wondering whether the authors could extend their analytical description of the replicator dynamics. For example, in simple cases like $Q=1$ or $Q=2$, is it possible to calculate the exact position of the various fixed points? In particular, it would be interesting to know how different types of errors affect the position of the fixed point between ALLD and DISC.

(-) It seems to me some of the results are driven by some implicit assumptions.

For example, while establishing an institution solves the coordination problem between individuals, new coordination problems might arise if there is more than one institution. While I understand that the present study does not explore this issue (it is assumed that there can be at most one institution), I would appreciate if the authors could discuss this issue.

As another example, to be able to explore the evolution of institutions in the first place, the model requires that there are Q predetermined individuals who would collectively form the institution. Also here, I could easily imagine another coordination problem to appear (for example, if different population members differ in whose opinion they trust).

Finally, the model only allows for three or four particular strategies (ALLD, ALLC, DISC; sometimes also reverse DISC). While this is a pretty standard assumption in the field, it makes it somewhat difficult to assess how robust the results are if other strategies were allowed. For example, one reason why image scoring has been suspected to be unstable is that individuals may have no incentive to have a score that is much better than necessary. I could well imagine that a similar idea applies to all tolerant institutions. For example, if a player knows it is sufficient to have a good reputation in the eyes of one of $Q=50$ institution members, and if institution members form their opinions independently, it might be a profitable deviation from DISC to only adopt DISC in 10% of all interactions, and to adopt ALLD in all others. I would appreciate if the authors could address this possible weakness.

(-) It would be easier to understand the authors' approach if they provided more details in the methods section. For example, at least for the first two equations (1) and (2), I'd appreciate if the authors could explain the logic of these equations.

They should also explain in a bit more detail how these four equations in g_X , g_Y , g_Z and g are solved (for example, is it obvious that there is always a unique solution for g in $[0,1]$, for any Q ?)
Finally, I'd appreciate if the methods section explains more formally how empathy is incorporated (without referring to the authors' previous paper).

Minor comments.

(-) It would be great if the authors could provide some explicit real world examples of the kind of institutions they have in mind. Credit bureaus, maybe?

(-) In the context of stochastic evolution in finite populations, the word "basin of attraction" is somewhat misleading (it is usually only used for differential equations). Maybe "expected fixation probability"?

(-) In the final paragraph, the authors say that it is a standard assumption in the literature that norms are universally accepted. I somewhat disagree. At least I think it is worth to mention studies that allow for different norms to compete, like Uchida and Sigmund (JTB 2010) or Yamamoto et al (SciRep 2017).

(-) On page 4, please explain how the 2x2 matrix notation for different social norms needs to be interpreted.

(-) On page 15, I think it should read "designated players" instead of "designed players"

Reviewer #2 (Remarks to the Author):

The authors carried out an evolutionary game analysis of indirect reciprocity (i.e., cooperation in the donation game sustained by a reputation mechanism). The topic has been mathematically investigated for more than 20 years. There have been proposed various mechanisms supporting reputation-based indirect reciprocity. However, how reputation is managed and propagates to the population has been relatively unclear despite various papers in the past 15 years or so. The authors analyzed the case in which public institutions (i.e., institutions that issue the reputation of each player that is shared by all the players) and private reputations (i.e., reputation of player A depends on an observer. So, player B may think player A is good, while player C may think player A is bad) compete. They showed that public institutions can evolve when they compete with private reputations. How the private versus public reputation mechanisms and different variations of them may evolve is a new vista to the study of indirect reciprocity. Furthermore, the results are both promising and surprising: They found that (i) public institutions can evolve in generous conditions and maintain cooperation, and (ii) a simple social norm (called scoring), which was initially proposed by Nowak-Sigmund, Nature 1998 and later discovered not to promote cooperation (Ohtsuki and Iwasa, Journal of Theoretical Biology 2004 and various subsequent studies) can promote cooperation under the present model. I personally see that

these two are main results and both are novel and relevant to understanding cooperation in our society. The paper is well written and the analyses are sound, while I have some comments which I believe that the authors should address. Therefore, I recommend the publication of this work in Nat Comm after the authors address my concerns. My specific feedback is as follows.

[major]

p.4, last paragraph: The public institution has been studied for a long time in the sense that, unless private reputations are explicitly modeled and studied, reputation-based indirect reciprocity papers implicitly assume public institutions that maintain and issue the reputation of each player (starting from e.g. Nowak-Sigmund 1998). The public institution that the authors model is much more involved than a usual public institution model. It consists of N assessors and a quorum with a threshold q is taken. I understand that the authors wanted to introduce a threshold to distinguish between generous institutions and strict institutions. Their work itself is good, looking at dependence of the results on q . However, their model of public institution is new and something different from common assumptions. I want to see more justification, including the comparison with past models of public institutions in reputation-based indirect reciprocity.

p.8, section "Evolution of public institutions ...": The analysis in this section is weaker in the previous sections. For example, in the previous sections, the authors essentially considered a full set of initial conditions. However, in this section, it seems that a fairly limited situations is considered. The authors considered a population consisting of DISC players as initial conditions for this analysis (second paragraph of this section). What if the different strategiests are iitially mixed? Furthermore, they did invasion analysis only. Is it possible to set up replicator dynamics, and run it even numerically to "semi-analytically" (semi means replicator equation itself is analytically constructed, so it is more analytical than mere agent-based simulations) support the conclusions? This may address a wider initial condition. Also, what if the population starts with half players using private information and the other half using public institutions? By saying these, I am not requesting a full-blown analysis. However, the authors should show some more evidence to indicate that their results in this section hold under sufficiently general conditions, not in particular conditions (particularly in terms of initial conditions) that are assumed in the current manuscript.

[minor]

p.2, paragraph beginning with "One way to": I think the authors can turn this more positive. The current text reads as if: the authors proposed an emphathy mechanism in their previous work. Some nice work done and it was in fact published in a good venue. Then, in this paragraph, the authors are saying as if what they did in their previous paper was not good therefore one needs a different explanation. This is as if the authors are disregarding their own work published recently in a good venue. I think which one (i.e., empathy or public institutions) is more realistic depends on different factors, such as the population size and particular social situations. I suggest that the authors rephrase this paragraph to give more positive credits to their previous work.

p.4, second paragraph in the Model section: The authors say that three strategies are considered. However, in the SI, I find four strategies (i.e., three plus the reverse DISC).

p.7, paragraph beginning with "By contrast to" and the paragraph that follows: These paragraphs describe Figure 3. I believe that the conclusions stated are also supported by Figure 2. It should be nice if there is some text mentioning to Figure 2, to better connect the infinite population and finite population analyses.

p.10, middle: "a intuition" -> "an intuition"

p.10, line up 11: "uniformly random initial frequency". This is vague. I do not think that the frequency itself is uniformly at random (what does it mean anyways? Dirichlet distribution?). Reading the SI, I fathomed that the authors selected the strategy of each player uniformly at random. Then, in the limit of large population, the frequency of each of the three strategies (let say) approaches 1/3. So, I would not call it "uniformly at random". Please explain the initial condition precisely. As a side note, by measuring the size of the attractive basin in the replicator dynamics in the previous section, the authors are implicitly assuming a Dirichlet-like distribution. It is different from the initial condition with which each player selects one of all the possible strategies with the equal probability. I am not saying that the initial condition the authors selected is bad. But, the initial condition should be stated clearly.

Reviewer #3 (Remarks to the Author):

Review of Radzvilavicius and Plotkin, Evolving public institutions that foster cooperation

In this paper the authors extend their previous work on endogenous reputation formation to allow a given fixed number of actors to make reputation judgments. R&P call this an "institution." They show that payoff biased cultural transmission will lead to the spread of a tendency to rely on institutional rather and personal judgments. They also show that such reliance is more likely to evolve when personal judgments are more self-interested and less empathetic.

This is an interesting paper with worthwhile results. However, I think the claims made are much too broad both because the model avoids a crucial issue that stands in the way of the formation of institutions and because it applies only to two person interactions and so does not explain much cooperation supported by norms, even in the simplest societies. The authors need to rewrite to paper to clarify the limits of their model as detailed below. I also think that their model makes a very interesting prediction that should be emphasized more. Such a paper would be a contribution to the theory of indirect reciprocity. Whether it is of sufficient interest to be suitable for publication in Nature Communications I leave to the editors.

The paper suggests that it explains the evolution of institutions, but what is actually explained is why people pay attention to institutional judgements. This seems like the easy part to me. Individuals who can coordinate their moral judgements with others do better because they are less likely to be punished because they cooperate when their next partner thinks they should have defected. Under the right

circumstances, R&P show that groups of 2 of 50 people who aggregate information and make it available, serve this function, so the tendency to adhere to their judgments spreads. The basic intuition is simple, and the real contribution of the paper is showing when it is true. They don't address the much harder question of why people would participate in such institutions, and if so, why they should do so honestly. Suppose $Q=2$, two judges named Joe and Jane. Joe and Jane are making judgments about the reputations of others. If they say Jill is bad, Jill suffers. Jill is not going to like this and will tend to impose costs on the Joe and Jane. So who needs this kind of trouble? Alternatively, Joe and Jane are going to be tempted to ask Jill for a bribe. What's to stop them? Empirically the answer is that judges are granted status and dishonesty is punished. It's easy to see why such norms provide a group benefit, but not at all obvious that they can be favored by individual natural selection. I don't think publication should be made contingent on solving this harder problem, but do think it should be made contingent on being much clearer that they have explained why people might pay attention to institutional judgments, once honest institutions have been established.

I also think that the paper should make more of the finding that empathy and institutions are substitutes. There is a sizable literature in economics about the crowding out phenomenon. People's intrinsic moral motivations are "crowded out" by more formal institutions. This paper suggests that the tendency to attend to institutions is most likely to evolve when their isn't much empathy which in this model stands in for personal moral motivations. Here people have types, while in real life the same person switches from intrinsic to formal, but still this is a cool result and should be foregrounded. Throughout the paper claims to apply to "large-scale" cooperation. By this the authors mean cooperation that is widespread in a population. Most people use large scale cooperation to apply to cooperation among large groups of people. There is good evidence that even in foraging societies people regularly cooperate in groups of hundreds of people---in communal hunting, construction of shared facilities, and in warfare. Explaining such large scale cooperation in an organism with primate reproductive biology is an important problem, and there is much debate about the mechanisms involved. Explaining pairwise cooperation is much easier because much less assortment is necessary to get cooperation to evolve in the first place, and because reciprocal strategies for pairs are less sensitive to errors than in larger scale cooperation. The paper must make this difference clear.

Here are some more detailed comments keyed to page/paragraph

2/4 Unclear. Is empathy favored by selection at the individual level without assortment?

3/1 Don;t see where the mean comes in here.

4/4 The assumption seems to be that observers can observe every act of every individual in the population. If so, say so.

4/5 N has not been defined that I can see.

5/2 An odd definition. Why exclude $q=1$? Some words would be helpful too. Why not just say everybody but one must agree.

5/4 Are we assuming that these two guys can observe every act of every individual? Lot's of stuff

happens indoors, out in the forest, etc. This assumption is equivalent to assuming this is a society without secrets. Not likely in my view.

5/4 The definition of strict thresholds requires that $1/2 < q < 1$, no? So doesn't this mean that both must agree that she has a good reputation? Seems like a complicated, long-handled way to say something very simple.

8/6 This is reminiscent of the crowding out phenomena. Seems like an important prediction.

Reviewer #1 (Remarks to the Author):

Summary.

The paper explores the impact of institutions on the evolution of indirect reciprocity. Indirect reciprocity is one of the major mechanisms to explain why humans may sometimes help complete strangers. The idea is that by helping someone, people increase their reputation. A good reputation, in turn, makes it more likely to receive cooperation in future.

A serious limitation of indirect reciprocity, however, seems to be that it makes rather stringent assumptions on how individuals share information. In the classical model of Ohtsuki and Iwasa it was shown that cooperation can evolve if all information is disseminated by a central observer. One consequence of this assumption is that all individuals agree on which reputation should be assigned to any given population member. In contrast, in a few recent studies it was shown that cooperation has problems to evolve if population members form their opinions independently. In that case, slight disagreements between players can quickly proliferate. In the long run, these disagreements can lead to the breakdown of cooperation.

In this present study, the authors connect these two extremes of public and private information. Using a mixture of analytical methods and simulations, the paper explores how public information sharing might evolve in the first place. Once such central institutions are in place, cooperation can evolve more easily than in a completely private setup. In fact, under some circumstances cooperation can prevail even for social norms that have been previously thought to be insufficient for the evolution of stable cooperation (Scoring and Shunning).

General evaluation.

The paper makes a very interesting contribution to the field. It provides a possibility to reconcile the findings of Ohtsuki and Iwasa with more recent work on private information. The used methods are sound and the main text is a pleasure to read.

Thank you for this careful reading of our paper and summary of its results in the context of prior work. We are especially in debt to the referee for the constructive comments below, which have led us to undertake new research and re-writing.

Having said that, there are also a few places where I think the manuscript can be improved. For example, while the model depends on quite a few parameters, only two parameters (q and Q) are systematically varied throughout the study. This makes it somewhat difficult to assess how general and robust the shown results are. In addition, I believe the discussion section would benefit from a more critical evaluation of some of the models' assumptions. Finally, I would like to encourage the authors to provide more detailed information in their methods section.

Once these points I addressed, I think the paper makes a very welcome contribution to the field. Such a paper is certainly worth being published in Nature Communications. Please find more details below.

We have thoroughly expanded our research, adding both new simulations and new mathematical analysis. This new work has allowed us to characterize the generality of our results and the relative importance of parameters, as described below.

Major suggestions.

(-) Robustness: The model depends on a number of parameters. For example, already the simple baseline model depends on the benefit of cooperation, and on different error rates. The stochastic model additionally depends on the mutation rate, the selection strength, and population size. I would appreciate to see some indication on whether the presented results are robust with respect to changes in these parameters. Also, what is the qualitative effect of these parameters? For example, is the outcome the same if assessment errors occur at a much higher rate than implementation errors?

As the referee suggests we have undertaken extensive analysis of parameter dependence.

The most important point is the relative impact of error rates (e_1 and e_2) versus the selection parameters (b and c). The take-away message, which we describe in the revision, is that errors have much smaller effects on strategy dynamics and equilibria than does selection. Likewise, population size and stochastic effects do not drive our qualitative results – and, indeed, the behavior under finite N can be predicted from the (deterministic) gradient of selection alone, as described below.

To make these points clear in the revised manuscript we now include phase portraits for replicator dynamics in three different regimes of error rates: $e_1 \gg e_2$, $e_1=e_2$, and $e_1 \ll e_2$ (Figure 3, Supplementary Figures 10 and 11). All three regimes exhibit very similar dynamics, because the selection terms dominate the effects of errors. We also show phase portraits of replicator dynamics for $b/c = 10$ (Supplementary Figure 12), which are visibly different than for $b/c=5$, highlighting the primary role of selection in these dynamics.

We have also systematically explored the impact of benefits and costs (b and c) on evolutionary dynamics. In particular, as the revised Methods section explains, strategy dynamics depend primarily on the ratio b/c . The revised work includes a new figure (calculated analytically) that shows the effect of b/c , and errors e_1 and e_2 , on the dynamics of institutional adherence, for each of the four social norms and for both strict and tolerant institutions (Figure 5).

While performing these analyses we discovered one unexpected prediction that we would not have been able to uncover without undertaking this new mathematical analysis: adherence to a tolerant institution is expected to selectively invade under Stern Judging provided b/c is large (>50), whereas otherwise adherence to Stern Judging invades only for a strict institution (Figure 5). We verified this analytical prediction in new Monte Carlo simulations, which we present in the revised manuscript (Supplementary Figure 8). We thank the referee for encouraging us to systematically study the effects of these parameters.

In general I was wondering whether the authors could extend their analytical description of the replicator dynamics. For example, in simple cases like Q=1 or Q=2, is it possible to calculate the exact position of the various fixed points? In particular, it would be interesting to know how different types of errors affect the position of the fixed point between ALLD and DISC.

Thank you for this suggestion. We have expanded our description of the replicator equations, and how they are derived.

We can indeed find closed-form analytic expressions for equilibria, for Q=1 and Q=2. Under the norm Simple Standing, for example, we can express the unstable equilibrium along the ALLD-DISC edge as the solution (by radicals) to a polynomial equation. In the case of Q=1 (and without empathy), the equilibrium is X=0, Z=1-Y where Y satisfies:

$$Y = \frac{b^2(e_1 - 1)(2e_2^2 - 3e_2 + 1) - 2bc(e_2 - 1) - c^2}{(e_1 - 1)(2e_2^2 - 3e_2 + 1)(b^2 - c^2)}$$

Whereas a strict institution with Q=2 produces an equilibrium along the ALLD-DISC edge at X=0, Z=1-Y, where Y satisfies:

$$Y = \frac{1}{2} \left(\frac{2b^2(e_1 - 1)(2e_2^2 - 3e_2 + 1)(e_1^2(2e_2^2 - 3e_2 + 1) + e_1(-4e_2^2 + 6e_2 - 4) + 2e_2^2 - 3e_2 - 1) + bc(4e_1^2(2e_2^2 - 3e_2 + 1) + e_1(-4e_2^2 + 6e_2 - 3) - 4e_2^2 + 6e_2 - 3) + c^2(e_1 + 1)}{b(e_1 - 1)(2e_2^2 - 3e_2 + 1)(b(e_1^2(2e_2^2 - 3e_2 + 1) + e_1(-4e_2^2 + 6e_2 - 4) + 2e_2^2 - 3e_2 - 1) + 2c(e_1 + 1))} \right. \\ \left. - \sqrt{\frac{c^2(e_1 + 1)^2(b^2(8e_2^2 - 12e_2 + 5) - 2bc(4e_2^2 - 6e_2 + 3) + c^2)}{b^2(e_1 - 1)^2(2e_2^2 - 3e_2 + 1)^2(b(e_1^2(2e_2^2 - 3e_2 + 1) + e_1(-4e_2^2 + 6e_2 - 4) + 2e_2^2 - 3e_2 - 1) + 2c(e_1 + 1))^2}} \right)$$

And finally a tolerant institution with Q=2 produces an equilibrium along the ALLD-DISC edge at X=0, Z=1-Y, where Y satisfies:

$$Y = \frac{1}{2} \left(\frac{2b^2(e_1 - 1)(2e_2 - 1)(e_1^2(2e_2^3 - e_2^2 - 2e_2 + 1) - 2e_1(2e_2^3 - e_2^2 + e_2 - 1) + 2e_2^3 - e_2^2 + 1) - bc(4e_1^2e_2(2e_2 - 1) + e_1(-4e_2^2 + 2e_2 + 1) - 4e_2^2 + 2e_2 + 1) + c^2(e_1 + 1)}{b(e_1 - 1)(2e_2 - 1)(b(e_1^2(2e_2^3 - e_2^2 - 2e_2 + 1) - 2e_1(2e_2^3 - e_2^2 + e_2 - 1) + 2e_2^3 - e_2^2 + 1) - 2c(e_1 + 1)e_2)} \right. \\ \left. - \sqrt{\frac{c^2(e_1 + 1)^2(b^2(8e_2^2 - 12e_2 + 5) + 2bc(4e_2^2 - 2e_2 - 1) + c^2)}{b^2(e_1 - 1)^2(1 - 2e_2)^2(b(e_1^2(2e_2^3 - e_2^2 - 2e_2 + 1) - 2e_1(2e_2^3 - e_2^2 + e_2 - 1) + 2e_2^3 - e_2^2 + 1) - 2c(e_1 + 1)e_2)^2}} \right)$$

In general, for all four norms, with Q=2 we can express solutions for all equilibria as solutions by radicals to polynomials of degree up to 4. Nonetheless, we do not include these unwieldy expressions in the revised manuscript, because we do not feel they provide much intuition or insight. The more important point, which the referee raised above, is the relative impact of error rates e_1 and e_2 versus the selection parameters b and c , as described in the revised manuscript.

(-) It seems to me some of the results are driven by some implicit assumptions.

For example, while establishing an institution solves the coordination problem between individuals, new coordination problems might arise if there is more than one institution. While I

understand that the present study does not explore this issue (it is assumed that there can be at most one institution), I would appreciate if the authors could discuss this issue.

As another example, to be able to explore the evolution of institutions in the first place, the model requires that there are Q predetermined individuals who would collectively form the institution. Also here, I could easily imagine another coordination problem to appear (for example, if different population members differ in whose opinion they trust).

We entirely agree: our work addresses how *adherence* to a public institution will influence levels of cooperation, and whether adherence will spread in a population that does not initially adhere. We have *not* studied how the public institution itself is assembled, or the coordination problem of multiple competing institutions. We have clarified this distinction explicitly in the revised manuscript (Discussion paragraph 6).

In reality, an institution of public monitoring is typically generated top-down – eg by constitutional agreement (in the case of governance), by law (in the case of credit bureaus), or by corporations (in the case of e-commerce sites, such as eBay). Although the problem of institution formation is outside the scope our analysis (it is really a question for political science), we mention these points in the revised Discussion section. Moreover, we add some analytical insights into the level of taxation that a society would rationally accept to support the formation and operation of a public institution of monitoring.

Finally, the model only allows for three or four particular strategies (ALLD, ALLC, DISC; sometimes also reverse DISC). While this is a pretty standard assumption in the field, it makes it somewhat difficult to assess how robust the results are if other strategies were allowed. For example, one reason why image scoring has been suspected to be unstable is that individuals may have no incentive to have a score that is much better than necessary. I could well imagine that a similar idea applies to all tolerant institutions. For example, if a player knows it is sufficient to have a good reputation in the eyes of one of $Q=50$ institution members, and if institution members form their opinions independently, it might be a profitable deviation from DISC to only adopt DISC in 10% of all interactions, and to adopt ALLD in all others. I would appreciate if the authors could address this possible weakness.

We agree that there is a high-dimensional space of mixed strategies, including the example mentioned, that we have not systematically explored. We describe this limitation, which is common to the field, in the revised text.

With respect to a more diverse strategy space, we have at least added a new mathematical analysis of a five-strategy mix containing ALLC, ALLD, and two types of DISC players: DISC players who adhere to the public institution, and DISC players who use their private views alone. As it turns out, analyzing all five types of strategies simultaneously is somewhat complicated, as it depends on whether the institution members themselves follow in the institution or not when forming their assessments (we have analyzed both situations, see

Methods Section 2). These new analyses expand the strategy space and add additional insights into effects of different types of institutions on cooperation.

(-) It would be easier to understand the authors' approach if they provided more details in the methods section. For example, at least for the first two equations (1) and (2), I'd appreciate if the authors could explain the logic of these equations.

Thank you for this suggestion. We have expanded the derivation and description of these equations (see Methods Sections 1 and 2), so that a reader can understand the work without reference to prior literature.

They should also explain in a bit more detail how these four equations in g_X , g_Y , g_Z and g are solved (for example, is it obvious that there is always a unique solution for g in $[0,1]$, for any Q ?) Finally, I'd appreciate if the methods section explains more formally how empathy is incorporated (without referring to the authors' previous paper).

For $Q \leq 2$, solutions can be expressed analytically as solutions to polynomials. For arbitrary Q , solutions can still be constructed by numerical iteration: we initially choose $\{g_X=g_Y=g_Z=0.5\}$, and then plug into equations for g_X , g_Y , g_Z to get successive iterates that converge to a unique solution. (The quantity g is a linear combination of g_X , g_Y , and g_Z and can be substituted into the equations for those three unknowns). This iterative method is guaranteed to converge to a unique solution in $[0,1]^3$ by Banach's fixed point theorem, because the form of the equations are convex combinations of the current iterate (g_X , g_Y , g_Z) with coefficients in the interior of the 3-simplex. We have described this procedure in the revised Methods section.

Minor comments.

(-) It would be great if the authors could provide some explicit real world examples of the kind of institutions they have in mind. Credit bureaus, maybe?

Thank you for this great suggestion. We have added this example, and a few others, to the start of the Discussion.

(-) In the context of stochastic evolution in finite populations, the word "basin of attraction" is somewhat misleading (it is usually only used for differential equations). Maybe "expected fixation probability"?

We agree, it was an abuse of terminology. We have fixed it.

(-) In the final paragraph, the authors say that it is a standard assumption in the literature that norms are universally accepted. I somewhat disagree. At least I think it is worth to mention studies that allow for different norms to compete, like Uchida and Sigmund (JTB 2010) or Yamamoto et al (SciRep 2017).

We agree, there are indeed a few papers that explore competing norms, and we now cite Uchida Sigmund (2010) and Yamamoto et al (2017), as well as discovery of new effective norms such as Staying (Sasaki 2017).

(-) On page 4, please explain how the 2x2 matrix notation for different social norms needs to be interpreted.

Thank you, we have revised this and expanded the Methods to make the mathematical presentation self-contained without reference to external literature for notational conventions

(-) On page 15, I think it should read "designated players" instead of "designed players"

Thank you, fixed.

Reviewer #2 (Remarks to the Author):

The authors carried out an evolutionary game analysis of indirect reciprocity (i.e., cooperation in the donation game sustained by a reputation mechanism). The topic has been mathematically investigated for more than 20 years. There have been proposed various mechanisms supporting reputation-based indirect reciprocity. However, how reputation is managed and propagates to the population has been relatively unclear despite various papers in the past 15 years or so. The authors analyzed the case in which public institutions (i.e., institutions that issue the reputation of each player that is shared by all the players) and private reputations (i.e., reputation of player A depends on an observer. So, player B may think player A is good, while player C may think player A is bad) compete. They showed that public institutions can evolve when they compete with private reputations. How the private versus public reputation mechanisms and different variations of them may evolve is a new vista to the study of indirect reciprocity. Furthermore, the results are both promising and surprising: They found that (i) public institutions can evolve in generous conditions and maintain cooperation, and (ii) a simple social norm (called scoring), which was initially proposed by Nowak-Sigmund, Nature 1998 and later discovered not to promote cooperation (Ohtsuki and Iwasa, Journal of Theoretical Biology 2004 and various subsequent studies) can promote cooperation under the present model. I personally see that these two are main results and both are novel and relevant to understanding cooperation in our society. The paper is well written and the analyses are sound, while I have some comments which I believe that the authors should address. Therefore, I recommend the publication of this work in Nat Comm after the authors address my concerns. My specific feedback is as follows.

Thank you very much for your detailed reading of the study and the very constructive comments below.

[major]

p.4, last paragraph: The public institution has been studied for a long time in the sense that, unless private reputations are explicitly modeled and studied, reputation-based indirect reciprocity papers implicitly assume public institutions that maintain and issue the reputation of each player (starting from e.g. Nowak-Sigmund 1998). The public institution that the authors model is much more involved than a usual public institution model. It consists of N assessors and a quorum with a threshold q is taken. I understand that the authors wanted to introduce a threshold to distinguish between generous institutions and strict institutions. Their work itself is good, looking at dependence of the results on q . However, their model of public institution is new and something different from common assumptions. I want to see more justification, including the comparison with past models of public institutions in reputation-based indirect reciprocity.

We agree – a simple institution provide public information is the standard model in the literature. This standard model supports cooperation under Stern Judging, but virtually no other norms.

Our goal was to study how more sophisticated institutions can expand range of cooperation across social norms. The type public institutions we study, with $Q > 1$ members, are realistic. In fact, institutions of this type are used in many real-life situations, such as reputation systems for sellers in e-commerce companies (eBay) that aggregate input from many user inputs. We explain this motivation in more detail in the revised manuscript, especially in the revised Discussion section that places our result in context.

The manuscript includes a comparison to the standard model of public information, which corresponds to the special case $Q=1$ in our model. We have made this comparison more explicit in the revised text. The main result is that, when institutional tolerance q is chosen appropriately, an institution with $Q > 1$ can support high levels of cooperation for *any* social norm, whereas the classic model of a public institution ($Q=1$) can support cooperation only for Stern Judging.

p.8, section "Evolution of public institutions ...": The analysis in this section is weaker in the previous sections. For example, in the previous sections, the authors essentially considered a full set of initial conditions. However, in this section, it seems that a fairly limited situations is considered. The authors considered a population consisting of DISC players as initial conditions for this analysis (second paragraph of this section). What if the different strategists are initially mixed?

Furthermore, they did invasion analysis only. Is it possible to set up replicator dynamics, and run it even numerically to "semi-analytically" (semi means replicator equation itself is analytically constructed, so it is more analytical than mere agent-based simulations) support the conclusions?

Thank you for encouraging us to strengthen the analysis of this section. We have added substantial new research, both simulation and mathematical analysis, to address the problem of a mixed population containing some individuals that follow the public institution and others who use their private assessment alone.

These additions include new simulation results that show the results of invasion of adherence starting from four different locations in the phase space, including an equal mix of DISC, ALLD, and ALLD strategies (Supplementary Figures 3, 4, 5, and 6).

More importantly, we have also followed the referee's suggestion to develop an analytic replicator-dynamic analysis of the problem of invasion of public monitoring. It is somewhat more complicated to study such a mixed population – when only a portion of individuals follow the public institution – because individuals may disagree about the reputation of a receiver. Nonetheless, we have developed the analytic replicator equation for this situation (Methods Section 2), and a new main text figure that shows effects of broad variation in error rates and game payoffs (Figure 5). This mathematical analysis in an infinite population provide accurate predictions for when (which norms, parameter values, etc) institution adherence will spread in

a finite population, as we verify by comparison to simulations (Supplementary Figures 7, 8, and 9)

In fact, in while developing this expanded analysis of a mixed population we discovered one unexpected prediction that would not have discoverable from simulations alone: adherence to a tolerant institution is expected to selectively invade under Stern Judging provided b/c is large (>50), whereas otherwise adherence to Stern Judging invades only for a strict institution. We verified this analytical prediction in new Monte Carlo simulations, which we present in the revised manuscript (Figure 4 and Supplementary Figure 8).

We thank the referee for encouraging us to mathematically analyze the case of a mixed population of adherents and non-adherents. The new analysis significantly expands to scope of our study.

This may address a wider initial condition. Also, what if the population starts with half players using private information and the other half using public institutions? By saying these, I am not requesting a full-blown analysis. However, the authors should show some more evidence to indicate that their results in this section hold under sufficiently general conditions, not in particular conditions (particularly in terms of initial conditions) that are assumed in the current manuscript.

Thank you again for this suggestion. We have included a complete mathematical analysis of the replicator dynamics for an arbitrary mix of ALLD, ALLC, DISC-private and DISC-public. And we included simulations showing fixation probabilities in populations containing such mixtures.

[minor]

p.2, paragraph beginning with "One way to": I think the authors can turn this more positive. The current text reads as if: the authors proposed an empathy mechanism in their previous work. Some nice work done and it was in fact published in a good venue. Then, in this paragraph, the authors are saying as if what they did in their previous paper was not good therefore one needs a different explanation. This is as if the authors are disregarding their own work published recently in a good venue. I think which one (i.e., empathy or public institutions) is more realistic depends on different factors, such as the population size and particular social situations. I suggest that the authors rephrase this paragraph to give more positive credits to their previous work.

Thank for this kind suggestion. We have modified the text.

p.4, second paragraph in the Model section: The authors say that three strategies are considered. However, in the SI, I find four strategies (i.e., three plus the reverse DISC).

Thank you for noticing this. We have removed RDISC from the analysis altogether. (RDISC only had any effect under Stern Judging, where it artificially removed the asymmetry between stern and tolerant institutions – effectively by re-interpreting “good” as “bad” and *vica versa*. And so we have removed it from the paper.)

p.7, paragraph beginning with "By contrast to" and the paragraph that follows: These paragraphs describe Figure 3. I believe that the conclusions stated are also supported by Figure 2. It should be nice if there is some text mentioning to Figure 2, to better connect the infinite population and finite population analyses.

Thank you for noticing this as well. We agree the infinite population analysis does a very good job at predicting the behavior in finite population We have added figures based on the new analysis of a mixed population containing both private- and public- follower (Figure 5, Supplementary Figure 8) that clarify that the infinite-population replicator dynamics accurately predict when adherence will invade or not in a finite population, described in the new section entitled “Replicator dynamics of institutional adherence”.

p.10, middle: "a intuition" -> "an intuition"

Thank you, fixed.

p.10, line up 11: "uniformly random initial frequency". This is vague. I do not think that the frequency itself is uniformly at random (what does it mean anyways? Dirichlet distribution?). Reading the SI, I fathomed that the authors selected the strategy of each player uniformly at random. Then, in the limit of large population, the frequency of each of the three strategies (let say) approaches 1/3. So, I would not call it "uniformly at random". Please explain the initial condition precisely. As a side note, by measuring the size of the attractive basin in the replicator dynamics in the previous section, the authors are implicitly assuming a Dirichlet-like distribution. It is different from the initial condition with which each player selects one of all the possible strategies with the equal probability. I am not saying that the initial condition the authors selected is bad. But, the initial condition should be stated clearly.

We agree, the wording was imprecise. We have clarified it in the revision (Methods Section 3).

Supplementary Figure 3 shows fixation probability of a single DISC-public mutant introduced into a population that initially is composed of ALLD, ALLC, and DISC-private each at frequency 1/3. Whereas Supplementary Figure 1 shows the *mean* fixation probability of DISC-public competing against DISC-private, where the initial frequency of DISC-public is drawn uniformly at random from the open interval (0,1), and the mean is taken over this distribution of initial frequencies.

Reviewer #3 (Remarks to the Author):

Review of Radzvilavicius and Plotkin, Evolving public institutions that foster cooperation
In this paper the authors extend their previous work on endogenous reputation formation to allow a given fixed number of actors to make reputation judgments. R&P call this an “institution.” They show that payoff biased cultural transmission will lead to the spread of a tendency to rely on institutional rather and personal judgments. They also show that such reliance is more likely to evolve when personal judgments are more self-interested and less empathetic.

This is an interesting paper with worthwhile results. However, I think the claims made are much too broad both because the model avoids a crucial issue that stands in the way of the formation of institutions and because it applies only to two person interactions and so does not explain much cooperation supported by norms, even in the simplest societies. The authors need to rewrite to paper to clarify the limits of their model as detailed below.

We have completely re-written the introduction and discussion to focus on the point the referee raises here: we are not studying the emergence of the institution of public monitoring itself, but rather whether a population will benefit from adhering to a public broadcast, and whether the tendency to adhere is selectively advantageous and will spread when rare.

We also clarify that our model for cooperation is based on the pairwise donation game, which is standard for the field of indirect reciprocity, but it does not address problems like collective action in an n -player public goods game. (The role of reputations in multiple-player games is largely unexplored in the literature on indirect reciprocity, because reputations seem more salient and likely to influence behavior in pairwise interactions. We mention n -player games as an open future direction in the revised Discussion).

I also think that their model makes a very interesting prediction that should be emphasized more. Such a paper would be a contribution to the theory of indirect reciprocity. Whether it is of sufficient interest to be suitable for publication in Nature Communications I leave to the editors.

Thank you. Thank you especially for drawing our attention to the predictions related to “crowding out” – which we have highlighted in a dedicated section in the revised manuscript.

The paper suggests that it explains the evolution of institutions, but what is actually explained is why people pay attention to institutional judgements. This seems like the easy part to me.

We agree. The study is about whether people will *adhere* to institutional judgements, and how best to design institutional judgments to facilitate cooperation, given the “social norm” of how individuals are assessed based on their actions. Some of these results are not entirely trivial – i.e. what level of tolerance in the institution is optimal, given the social norm. We also analyze whether adherence will spread or not, when initially rare in a population.

We agree that the problem of how to *establish* an institution to begin with is an entirely separate and more difficult question. (This question is more in the domain of political science than evolutionary game theory.) We have clarified the scope of our study accordingly, modifying the text throughout the Introduction and Discussion (especially in Discussion paragraph 6).

That being said, the revised Discussion does contain some comments on how an institution might be established. In reality, such institutions of public monitoring are typically generated top-down – eg by constitutional agreement (in the case of governance), by law (in the case of credit bureaus), or by companies (in the case of e-commerce sites, such as eBay). The Discussion now includes a few analytical insights into the level of taxation that a society would rationally accept to support the formation and operation of a public institution of monitoring. These simple calculations also describe what tolerable level of wealth inequity would make such an institution robust to corruption by side-deals.

Individuals who can coordinate their moral judgements with others do better because they are less likely to be punished because they cooperate when their next partner thinks they should have defected. Under the right circumstances, R&P show that groups of 2 of 50 people who aggregate information and make it available, serve this function, so the tendency to adhere to their judgments spreads. The basic intuition is simple, and the real contribution of the paper is showing when it is true.

We agree with this characterization of the work.

They don't address the much harder question of why people would participate in such institutions, and if so, why they should do so honestly. Suppose $Q=2$, two judges named Joe and Jane. Joe and Jane are making judgments about the reputations of others. If they say Jill is bad, Jill suffers. Jill is not going to like this and will tend to impose costs on the Joe and Jane. So who needs this kind of trouble? Alternatively, Joe and Jane are going to be tempted to ask Jill for a bribe. What's to stop them? Empirically the answer is that judges are granted status and dishonesty is punished. It's easy to see why such norms provide a group benefit, but not at all obvious that they can be favored by individual natural selection. I don't think publication should be made contingent on solving this harder problem, but do think it should be made contingent on being much clearer that they have explained why people might pay attention to institutional judgments, once honest institutions have been established.

Again, we entirely agree with these points. And we have delineated the scope of our analysis more clearly in the revised manuscript.

Nonetheless, in the Discussion we do at least add some rudimentary insights into the formation of a public institution itself. We discuss the level of taxation that would be accepted by individuals to fund a public institution of judgement. This simple calculation gives us some quantitative feeling for the level of institutional robustness against the type of side-deal

corruption that the referee raises here. We thank the referee for raising these very realistic concerns, which have sharpened the work's scope considerably.

I also think that the paper should make more of the finding that empathy and institutions are substitutes. There is a sizable literature in economics about the crowding out phenomenon. People's intrinsic moral motivations are "crowded out" by more formal institutions. This paper suggests that the tendency to attend to institutions is most likely to evolve when their isn't much empathy which in this model stands in for personal moral motivations. Here people have types, while in real life the same person switches from intrinsic to formal, but still this is a cool result and should be foregrounded.

Thank you for drawing our attention to this result and its relationship to the "crowding out" phenomenon in economics. We have highlighted this result in the revised manuscript, including a dedicated section on crowding out, and callout in the Discussion.

There is indeed a relationship to the crowding out phenomenon. Importantly, this suggests that reliance on a formal institution may be detrimental, over the longterm, as it relaxes any pressure to maintain internal mechanisms such as empathy. Our results generally support this notion, but they do not do so uniformly. For example, adherence to a strict public institution can selectively invade a population under the Stern Judging norm, whether or not the private assessors employ empathy. And adherence to a tolerant institution can also invade under the Shunning norm even when private assessors have empathy (especially for a large institution). The revised manuscript describes the connection to "crowding out" in general, as well as these few counterexamples.

Throughout the paper claims to apply to "large-scale" cooperation. By this the authors mean cooperation that is widespread in a population. Most people use large scale cooperation to apply to cooperation among large groups of people. There is good evidence that even in foraging societies people regularly cooperate in groups of hundreds of people---in communal hunting, construction of shared facilities, and in warfare. Explaining such large scale cooperation in an organism with primate reproductive biology is an important problem, and there is much debate about the mechanisms involved. Explaining pairwise cooperation is much easier because much less assortment is necessary to get cooperation to evolve in the first place, and because reciprocal strategies for pairs are less sensitive to errors than in larger scale cooperation. The paper must make this difference clear.

We have absolutely clarified this distinction in the revision. We have removed the term "large-scale" throughout the text, which, we agree, was used inaccurately. And the revised Discussion mentions n -player games an area for future research in indirect reciprocity.

Here are some more detailed comments keyed to page/paragraph

2/4 Unclear. Is empathy favored by selection at the individual level without assortment?

Yes, that is correct. We have previously shown the empathy is typically favored by selection at the individual level even without any assortment (by, eg, population structure). However, this was studied in a model in which empathy was effortless: there was no cost associated with the ability to take another person's perspective. We clarify this in the revised text.

3/1 Don;t see where the mean comes in here.

Each member of the institution may have a potentially different view of a focal individual, good (1) or bad (0). The mean reputation among the Q members of the institution is compared to the threshold q to determine the public broadcast of the focal individual's reputation. For $q < 1/Q$, for example, this is tantamount to saying that the public broadcast is good provided at least one member of the institution sees the focal individual as good (see below).

4/4 The assumption seems to be that observers can observe every act of every individual in the population. If so, say so.

That is almost correct. An individual i engages in N interactions each round. Each observer observes a random one of these N interactions, for each focal individual i who is being assessed by the observer. And so, observers observe N interactions each round, instead of $N*N$.

4/5 N has not been defined that I can see.

Thanks, fixed.

5/2 An odd definition. Why exclude $q=1$? Some words would be helpful too. Why not just say everybody but one must agree.

Thank you – that text was much wordier and more complicated than needed. In the revision we have stated this condition clearly, in English, before making a precise mathematical definition.

5/4 Are we assuming that these two guys can observe every act of every individual? Lot's of stuff happens indoors, out in the forest, etc. This assumption is equivalent to assuming this is a society without secrets. Not likely in my view.

We agree, to a large extent. We do not actually assume the each observer sees all $N * N$ interactions, but rather that each observer sees N interactions (one for each person to be assessed). We expect similar results to hold even when observers are allowed to sample only a (random) subset of individuals for judgement, each round, provided reputations still equilibrate quickly compared to strategy changes in the population.

5/4 The definition of strict thresholds requires that $1/2 < q < 1$, no? So doesn't this mean that both must agree that she has a good reputation? Seems like a complicated, long-handled way to say something very simple.

Yes, the wording was far more complicated than needed; we have clarified.

8/6 This is reminiscent of the crowding out phenomena. Seems like an important prediction.

Thanks again for mentioning this point, which we now highlight in a new section of the revised main text devoted to motivational crowding out.

REVIEWER COMMENTS

Reviewer #1 (Remarks to the Author):

The authors have taken all my suggestions into account. They have substantially revised the manuscript, clarifying the manuscript's scope and generalizing the manuscript's conclusions.

In particular, they spell out more clearly what assumptions the model makes, and how robust the findings are with respect to possible parameter changes. I think these changes have further improved a manuscript that has been very good to begin with.

Overall, I believe the paper makes a very valuable contribution to the field of indirect reciprocity. While public reputation systems are important for the evolution of cooperation (especially in the presence of noise), previous work did not address how such coordination can be achieved. This work makes an important step into this direction.

I just have one minor comment: In the revised discussion, the authors mention that members of an institution receive an income of approximately $N(b-c)/Q$. I did not understand the logic of this approximation. Is it assumed here that any net benefit of cooperation in the entire population is equally divided among the institution members? Or should this be interpreted as an upper bound of how much a member of the institution could possibly obtain for its services, if taxes were used? A clearer exposition would help.

Reviewer #2 (Remarks to the Author):

The authors carried out a substantive revision to address the issues raised by the reviewers. I consider the revision is satisfactory, and I recommend the publication as is.

Reviewer #3 (Remarks to the Author):

The paper is somewhat improved, but still implies that the results explain much more than they actually explain. For example the title suggests that the paper will explain how to design institutions that foster cooperation. What the math actually does is show that once honest institutions built exist, people will pay attention the judgements that the institutions makes. The math is correct but the introduction and discussion imply that the results explain more than they actually do.

The basic problem is that the norms that maintain institutions are n-person cooperation problems. Judge Joe cheats. The paepr says he will be punished but does not explain why anybody should incur the costs to do this. It is an example of the second order free rider problem in an n person interaction. Explaining how to design institutions requires the solution of this and many other similar problems. The author suggest that institutions arise through top down processes. But most human societies f up until

the last few thousand years lacked formal structures to generate such institutions. Norms that create important n-person cooperation exist and are shared at ethnic scales in societies without chiefs, or any other political structure on that scale. Indirect reciprocity is interesting, but by focusing on two player interactions it omits much of what is important about human morality.

The paper provides interesting new results about indirect reciprocity, but it only inches our understanding of morality and cooperative institutions forward a small amount. I do not think it should be published in its present form.

Reviewer #2 (Remarks to the Author):

The authors carried out a substantive revision to address the issues raised by the reviewers. I consider the revision is satisfactory, and I recommend the publication as is.

We thank the referee for the detailed suggestions, which substantially improved the study.

Reviewer #1 (Remarks to the Author):

The authors have taken all my suggestions into account. They have substantially revised the manuscript, clarifying the manuscript's scope and generalizing the manuscript's conclusions. In particular, they spell out more clearly what assumptions the model makes, and how robust the findings are with respect to possible parameter changes. I think these changes have further improved a manuscript that has been very good to begin with.

Overall, I believe the paper makes a very valuable contribution to the field of indirect reciprocity. While public reputation systems are important for the evolution of cooperation (especially in the presence of noise), previous work did not address how such coordination can be achieved. This work makes an important step into this direction.

We thank the referee for the detailed suggestions. To recap, we undertook the following revisions in response to the initial reports from referee #1 and #2:

- Undertook a systematic analysis of parameter dependence (Figure 3, S10, S11, S12)
- Developed replicator equations (Methods Section 2) that allow analytic predictions for when selection favors adherence (new Figure 5), verified by Monte Carlo simulations
- Added analysis of institutional-adherents competing against a more diverse strategy space
- Expanded our derivation of the replicator dynamic equations, and explained why there is always convergence to a unique solution for reputations
- Added discussion of related work on competing social norms

We are happy that referees #1 and #2 are satisfied with the resulting, revised work.

I just have one minor comment: In the revised discussion, the authors mention that members of an institution receive an income of approximately $N(b-c)/Q$. I did not understand the logic of this approximation. Is it assumed here that any net benefit of cooperation in the entire population is equally divided among the institution members? Or should this be interpreted as an upper bound of how much a member of the institution could possibly obtain for its services, if taxes were used? A clearer exposition would help.

We have now clarified this portion of the text. The referee's first interpretation was correct: each individual contributes (up to) the net benefit of cooperation as a tax to support the institution, whose total proceeds are divided equally among the Q members of the institution.

Reviewer #3 (Remarks to the Author):

We want to thank the referee #3 both for the initial round of comments -- which led to substantial revision, including a new section on how our results relate to the “crowding out” phenomenon in economics – and also for the second round of comments on the revised manuscript.

Before responding to the revised comments, we want to review the referee’s original assessment and critique of the original manuscript:

This is an interesting paper with worthwhile results. However, I think the claims made are much too broad because the model avoids a crucial issue that stands in the way of the formation of institutions. The paper suggests that it explains the evolution of institutions, but what is actually explained is why people pay attention to institutional judgements. The authors need to rewrite to paper to clarify the limits of their model.

...

I think that their model makes a very interesting prediction that should be emphasized more. Such a paper would be a contribution to the theory of indirect reciprocity. Whether it is of sufficient interest to be suitable for publication in Nature Communications I leave to the editors. [Referee #3, initial report]

Referee #3 was absolutely correct: our study analyzes when individuals will *adhere* to a public institution of behavioral monitoring, as opposed to how an institution is formed. In response to this initial critique, we worked to limit the scope of our claims throughout the text.

Now we turn to the referee’s comments on the revised manuscript:

The paper is somewhat improved, but still implies that the results explain much more than they actually explain. For example the title suggests that the paper will explain how to design institutions that foster cooperation. What the math actually does is show that once honest institutions built exist, people will pay attention the judgements that the institutions makes. The math is correct but the introduction and discussion imply that the results explain more than they actually do.

Once again, we completely agree with this point. The paper explains why individual will adhere to a public institution of assessment – which has been a key, but hitherto unjustified assumption in the field of indirect reciprocity. We do *not* analyze how an institution is formed to begin with.

And so, in the second revision we have been fastidious in making absolutely sure that our claims are limited to the question of whether or not individuals will pay attention to the judgements of a public institution of reputation assessment.

We have changed the title of the paper to emphasize adherence as the topic of study: ***Adherence to public institutions that foster cooperation***.

We have made further modifications throughout the text, many of them at the explicit direction of the Editors, to limit the scope of our claims and to temper our conclusions. Some of the most important changes are quoted here:

“Adherence to public institutions that foster cooperation” [Title]

“Here we study evolution of **adherence to public monitoring...**” [Abstract]

“**We conclude that adherence to public institutions of moral assessment can spread naturally,** and we determine the conditions under which institutions produce high rates of cooperation for each of the four social norms we study.” [Introduction]

“**Our study does not address how an institution of public monitoring is established to begin with. Instead, we have focused on whether, and when, adherence to an existing public broadcast is beneficial.** The question of establishing an institution *de novo* is perhaps more difficult -- and more properly a question in political science than in evolutionary game theory. In practice, modern societies that possess formal systems of governance typically form public institutions by legislation and taxation. **Although the problem of forming an institution is outside the scope of our study,** we can nonetheless provide some rudimentary insights by quantifying the level of taxation that rational individuals would willingly pay to support such an institution, in order to reap the rewards of cooperation it provides.” [Discussion]

The basic problem is that the norms that maintain institutions are n-person cooperation problems. Judge Joe cheats. The paper says he will be punished but does not explain why anybody should incur the costs to do this. It is an example of the second order free rider problem in an n person interaction. Explaining how to design institutions requires the solution of this and many other similar problems.

We have removed all claims and language about how to *design* an institution – which, we agree, is outside the scope of the theory of indirect reciprocity.

As the revised text clarifies, the paper does not claim that a corrupt institution member (Judge Joe) will be punished, at a cost born by anyone. Instead, the revised text highlights the second-order free rider problem as an important topic for any future research on the formation of institutions.

The author suggest that institutions arise through top down processes. But most human societies up until the last few thousand years lacked formal structures to generate such institutions. Norms that create important n-person cooperation exist and are shared at ethnic scales in societies without chiefs, or any other political structure on that scale. I

We state clearly that the formation of an institution is outside the scope of our analysis. Also, in the revised text we limit our Discussion to modern societies that possess formal structures of governance.

Indirect reciprocity is interesting, but by focusing on two player interactions it omits much of what is important about human morality.

We agree that there are many important aspects of human morality that fall outside the scope of indirect reciprocity, and that are therefore not addressed by our study.

The paper provides interesting new results about indirect reciprocity, but it only inches our understanding of morality and cooperative institutions forward a small amount. I do not think it should be published in its present form.

We are pleased the referee sees the paper as providing interesting new results about indirect reciprocity. This is precisely the objective of our study.

REVIEWERS' COMMENTS

Reviewer #1 (Remarks to the Author):

I stand by my previous assessment that this paper makes a very welcome contribution to the theory of indirect reciprocity. Previous models considered two somewhat extreme cases (the case of purely public information and purely private information, respectively). The current approach offers a framework to explore how a public information institution can emerge in a society in which reputations are initially assigned privately.

Certainly, the authors' approach has its limitations. The model does not address how institutions are formed to begin with, and it does not formally address how one can safeguard such institutions against corruption. However, the authors clearly state these limitations in the main text, which I find appropriate.

The fact that adherence to centralised institutions can evolve is not obvious, even in this simple model. To me, the instances in which these institutions do not evolve are almost as interesting as the instances in which they do. This point is illustrated in Figures 4 and 5. For example, although a tolerant institution that implements the Shunning norm is beneficial (Figure 3, where cooperation rates are high for small q), such institutions can have problems to evolve (Figures 4 and 5).

Overall, I believe the model makes a very good contribution to the field, and it is transparent about its underlying assumptions. Perhaps it should also be noted that the authors have put a substantial effort into the original paper and into the revisions. Their mathematical analysis is non-trivial. Therefore, I support publication.

Some typos:

- (-) Page 3: "the degree to it tolerates"  "the degree to which it tolerates"
- (-) Page 5: "reputation updates reach equilibrium"  "reputation assessments reach an equilibrium"
- (-) Page 5: "are faster those of strategic change"  "are faster than those of strategic change"
- (-) Page 8: "One possibility has do with"  "One possibility has to do with"
- (-) Page 11: "external interventions can under undermine"  "can undermine"

Reviewer #3 (Remarks to the Author):

The contribution of the paper are more clearly described in the current version and I think it is now acceptable.